# Extended-Synaptotagmin-1 and -2 control T cell signaling and function

Nathalia Benavides ⬤ & Claudio G Giraudo ⬤ ✉

## Abstract

Upon T-cell activation, the levels of the secondary messenger diacylglycerol (DAG) at the plasma membrane need to be controlled to ensure appropriate T-cell receptor signaling and T-cell functions. Extended-Synaptotagmins (E-Syts) are a family of inter-organelle lipid transport proteins that bridge the endoplasmic reticulum and the plasma membrane. In this study, we identify a novel regulatory mechanism of DAG-mediated signaling for T-cell effector functions based on E-Syt proteins. We demonstrate that E-Syts downmodulate T-cell receptor signaling, T-cell-mediated cytotoxicity, degranulation, and cytokine production by reducing plasma membrane levels of DAG. Mechanistically, E-Syt2 predominantly modulates DAG levels at the plasma membrane in resting-state T cells, while E-Syt1 and E-Syt2 negatively control T-cell receptor signaling upon stimulation. These results reveal a previously underappreciated role of E-Syts in regulating DAG dynamics in T-cell signaling.

**Keywords** Extended-Synaptotagmins; T Cells; Plasma Membrane; DAG; T-cell Receptor Signaling
**Subject Categories** Immunology; Membranes & Trafficking

## Introduction

Human T lymphocytes orchestrate the immune response through specific signaling pathways, thereby producing chemical messengers called cytokines and recognizing and destroying cancerous and virus-infected cells. These highly regulated processes require the formation of a tight contact area between the lymphocyte-target cells, known as the immunological synapse (IS), where the polarized secretion of cytokines and apoptosis-inducing proteins occurs (Le Floc'h et al, 2013; Negulescu et al, 1996; Zhang and Bevan, 2011). The IS membrane is a heterogeneous region where interactions between lipids and protein complexes relay information from extracellular receptors to induce global cellular changes (Gawden-Bone and Griffiths, 2019).

Diacylglycerol (DAG) is a key secondary messenger generated at the IS primarily from the hydrolysis of phosphatidylinositol-4,5-bisphosphate ($PI(4,5)P_2$) by the enzyme phospholipase C gamma (PLCγ) upon T-cell receptor (TCR) activation. The localized concentration of DAG in the plasma membrane (PM) of activated lymphocytes is responsible for polarizing the microtubule organizing center (MTOC) during IS formation and key signaling pathways such as nuclear factor-κB (NF-κB) and mitogen-activated protein kinase (MAPK). T cells express two diacylglycerol kinases (DGKs), DGKα and DGKζ, to downregulate DAG-mediated signals. DGKs phosphorylate DAG to yield phosphatidic acid, thus terminating the T-cell immune response (Baldanzi et al, 2016). Therefore, control of DAG at the PM plays a fundamental role in fine-tuning the levels of T-cell activation, signaling for cellular effector functions, and spatial organization of the IS (Chauveau et al, 2014; Guo et al, 2008; Koretzky, 2003; Topham and Epand, 2009). Despite strong evidence supporting the critical role of DAG in controlling T-cell functions, our understanding of how T cells regulate the dynamic transfer of lipids from one cellular compartment to another remains scant. Growing evidence from other cell types suggests an essential role of lipid transport proteins in modulating cellular lipid homeostasis without the need for membrane fusion through membrane contact sites (Henne Liou and Emr, 2015).

The Extended-Synaptotagmin (E-Syt) protein family consists of three isoforms, all of which are lipid transport proteins that tether the endoplasmic reticulum (ER) to the PM (Saheki and De Camilli, 2017). E-Syts are anchored to the ER via a hydrophobic hairpin insertion followed by a cytosolic synaptotagmin-like mitochondrial-lipid-binding-protein (SMP) domain and multiple C-terminal C2 domains. E-Syt2 and -3 are constitutively bound to the PM through their C2 domains, which interact with $PI(4,5)P_2$, whereas E-Syt1 requires an increase in cytosolic $Ca^{2+}$ for interaction with the PM. E-Syt1 and -2 can also form homo- and heterodimers in vitro through their SMP-domain interactions, and thus mediate the transport of DAG down the concentration gradient between artificial membranes (Giordano et al, 2013). However, the physiologic role of E-Syt proteins remains unclear. Whether E-Syts constitute a specialized mechanism to regulate DAG concentration at the PM of T cells is still unknown.

Here, we showed that E-Syt1 and E-Syt2, but not E-Syt3, are highly expressed in human T lymphocytes. We found that E-Syts downmodulate T-cell activation by clearing DAG from the PM. To test this hypothesis, we used a combination of biochemical and immunological approaches to investigate each step of the T-cell activation process, from early TCR activation kinetics to cell-mediated functions. We showed that in the absence of E-Syts, T cells are present in a higher activated state due to DAG

---

Department of Microbiology and Immunology—Sidney Kimmel Medical College—Thomas Jefferson University, Philadelphia, PA, USA. ✉E-mail: Claudio.Giraudo@jefferson.edu

accumulation at the PM. Our results further suggest that DAG modulation at the PM by E-Syt proteins contributes to T-cell regulation independently of DGK-mediated mechanisms.

# Results

## E-Syt1 is a negative regulator of lymphocyte-mediated cytotoxicity in primary CD8+ T cells

E-Syt proteins are expressed at high concentrations in mammalian tissues, in particular, E-Syt1 and E-Syt2 are highly expressed in the lung and spleen (Tremblay and Moss, 2016). To evaluate the expression of E-Syt1-3 in human cells, we performed western blot (WB) analysis in primary human peripheral blood mononuclear cells (PBMCs), purified CD8+ T cells, and cell lines such as Jurkat CD4+ T cells, YTS natural killer cells, and HeLa epithelial cells. In contrast to E-Syt1 and E-Syt2 which were expressed in all the cells tested, E-Syt3 was only detected in HeLa and YTS cells (Appendix Fig. S1A). In addition, data from a mass spectrometry analysis of the interactome of Syntaxin 11 (STX11), a PM lipid-anchored fusion protein that mediates lytic granule exocytosis at the immune synapse (Bryceson et al, 2007; Spessott et al, 2017b; zur Stadt et al, 2005), showed that E-Syt1 specifically co-immunoprecipitated with GFP-STX11, but not with the control GFP alone in stimulated primary human CD8+ T cells (Appendix Fig. S1B). These observations prompted us to further explore the role of E-Syt1 and E-Syt2 in human T cells.

To investigate the role of E-Syt proteins in lymphocyte-mediated effector functions, we transfected human CD8+ T cells from healthy donors with small interfering RNAs (siRNAs) to downregulate the expression levels of endogenous E-Syt1, E-Syt2, and Munc13-4, an essential protein for cytolytic granule release (Feldmann et al, 2003). Knockdown efficiency was tested by WB (Fig. 1A). We first evaluated whether the mobilization of lytic granules was impaired by measuring the induced surface expression of CD107a on CD8+ T cells upon conjugation with P815 target cells. For this, siRNA-treated cells were incubated either alone (unstimulated) or with P815 target cells coated with human anti-CD3 (stimulated) to measure CD107a on the cell surface by flow cytometry (Benavides et al, 2020; Spessott et al, 2015). To properly investigate the different stages of IS formation, we tested a wide range of activating time periods (30–150 min). Measurements of CD107a on the cell surface revealed a significant increase in degranulation in cells with downregulated E-Syt1 and -2 proteins compared to control cells (Fig. 1B). Notably, this increase was predominantly observed at the early activation times: 30 and 60 min. In contrast, Munc13-4 knockdown cells displayed a significant reduction in granule release as previously described in both patient cells with Munc13-4 deficiency and Munc13-4 knockdown cells (Feldmann et al, 2003). Interestingly, E-Syt1&2 double-knockdown did not produce an additive effect. Together, these results suggest that both E-Syt1 and E-Syt2 independently modulate the efficiency of vesicle fusion and thus degranulation in CD8+ T cells.

To further investigate if the increased CD107a mobilization in E-Syt knockdown cells also influence with cytotoxic T lymphocyte (CTL) lytic activity, we measured lymphocyte-mediated cytotoxicity. We used a highly sensitive imaging-based assay that specifically monitors apoptosis over time by measuring the activation of Caspase-3 and Caspase-7 in single-target cells (Benavides et al, 2020). The results showed that upon E-Syt1 or E-Syt1&2 knockdown, a greater percentage of target cells died over time (Fig. 1C). This difference was evident when focusing on the rate of killing during the first one hundred minutes, as well as the total level of target cell death by the end of the 4 h. However, E-Syt2 knockdown CD8+ T cells displayed only a slight increase in target cell death whereas Munc13-4 knockdown cells showed a drastic reduction when compared with NT-siRNA control cells (Fig. 1C). Altogether, these results demonstrate that E-Syt1 alone and together with E-Syt2 negatively modulates CTL granule release and target cell killing.

Given the established role of E-Syt proteins in glycerophospholipid transfer and the role of DAG in T-cell responses, we evaluated whether E-Syt proteins influence either proximal TCR signaling or downstream pathways. We wanted to see if this functional change was due to increased signaling kinetics, therefore we activated E-Syt knockdown human PBMCs with anti-CD3/CD28 coated beads to mimic TCR engagement. The kinetics of the phosphorylation events of the molecules ZAP70, LAT, and PLCγ1, which are involved in early TCR signaling upstream of DAG formation, were analyzed by flow cytometry. Results showed a significant increase in the p-ZAP70 and p-LAT signaling kinetics in both CD4+ and CD8+ T-cell subtypes (Figs. 1D–F and EV1).

## E-Syt1 and E-Syt2 regulate TCR signaling kinetics

To rule out any off-target effect of the siRNAs, or variability in the response due to a heterogenous population of siRNA-treated cells, we generated *ESYT1*, *ESYT2*, and *ESYT1 &2* double knockout (DKO) Jurkat cells, using the CRISPR/Cas9 system and single-cell cloning isolation (Fig. 2A). Cells were activated and tested for signaling capabilities as described above. E-Syt2 KO and DKO cells showed a statistically significant increase in the rate (1–5 min) and the amount of p-ZAP70 (Fig. 2B) and pLAT (Fig. 2C) over time compared to control WT and E-Syt1 KO cells. The analysis of p-PLCγ1 kinetics showed only a slight increase in E-Syt2 KO and DKO cells at later time points (10–60 min) although the difference was not statistically significant when compared with WT and E-Syt1 KO cells (Fig. 2D). Absolute mean fluorescence intensity (MFI) values did not differ between control and E-Syt KO Jurkat cells at baseline before stimulation (time=0) (Appendix Fig. S2E–G). These results indicate that the absence of E-Syt2, but not of E-Syt1 alone in Jurkat cells leads to enhanced phosphorylation of upstream signaling proteins upon activation.

To further investigate the role of E-Syts in TCR signaling downstream of DAG formation, we evaluated the activation of the MAPK pathway. Using WB techniques, we compared phosphorylation levels of MAPK (p44/42) in E-Syt KOs and WT-Jurkat cells over time. Quantification of the p-MAPK bands showed that E-Syt2 KO cells displayed a significant increase in the amount of p-MAPK after CD3/28 stimulation compared to WT-Jurkat cells (Fig. 2E,F). In addition, quantification analysis revealed a two-fold increase in p-MAPK signal in E-Syt2 KO cells compared to WT-Jurkat cells before stimulation (0 min). Similar tests were performed using E-Syt1 and E-Syt1 & 2 DKO cells displaying an upward trend for both at early time points (Fig. 2F). Similarly, we observed increased signaling kinetics when Jurkat cells were treated with different

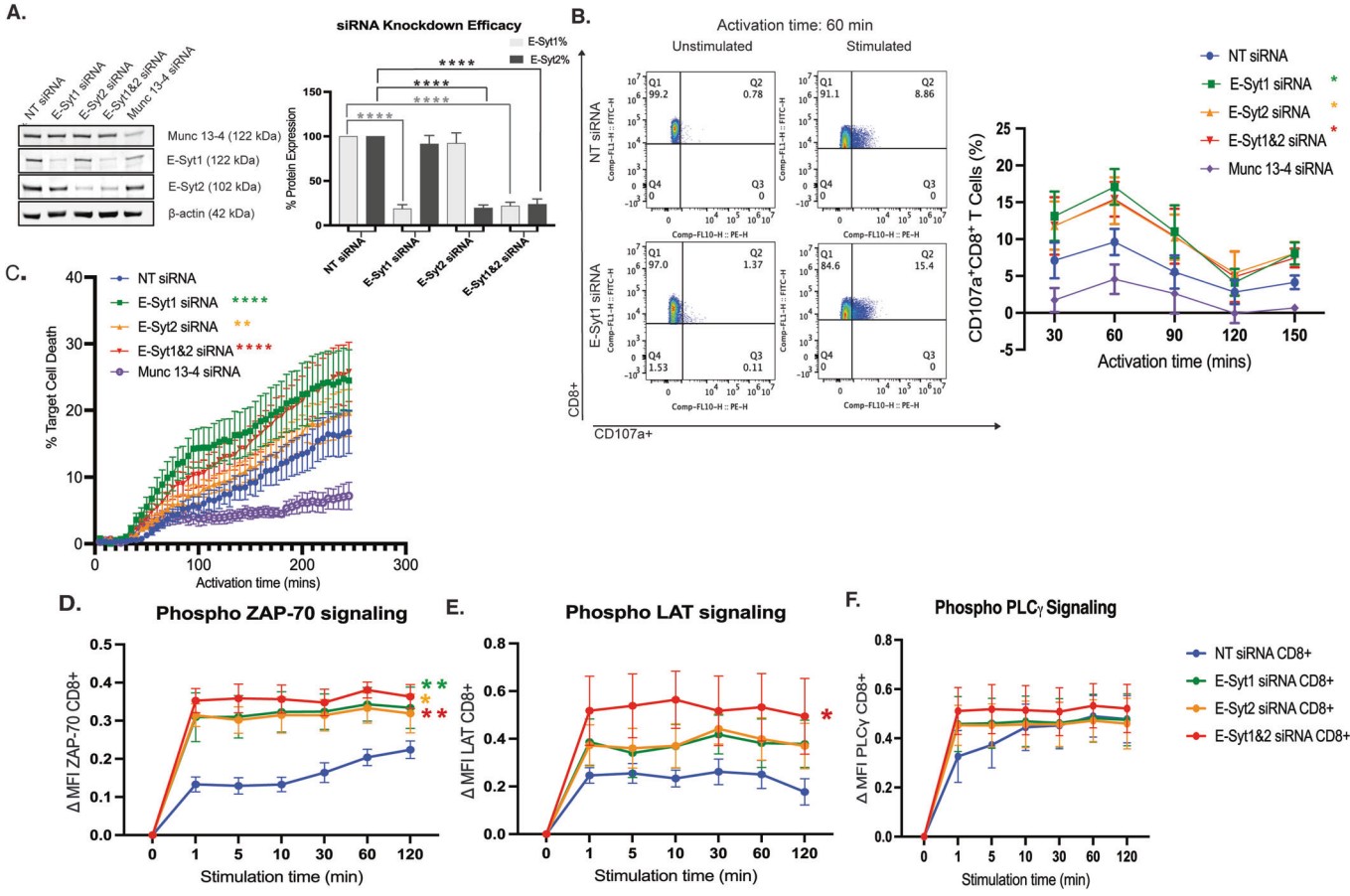

**Figure 1. E-Syt1 and -2 knockdown in primary CD8+ T cells enhances cytotoxic functions through increased TCR signaling.**

(A) Western blot analysis and quantification of cell lysates from human CD8+ T cells transfected with either non-targeting (NT), E-Syt1, E-Syt2, E-Syt1&2, or Munc13-4 siRNA. The data shown represent the mean ± SEM of at least five independent experiments. Significance was determined by one-way ANOVA with Bonferroni's multiple comparisons test against the respective E-Syt protein levels in control NT siRNA (****$P < 0.0001$). (B) CD107a degranulation assay of siRNA-treated human CD8+ T cells incubated in the absence (unstimulated) or the presence of P815 target cells coated with human anti-CD3 antibody (stimulated) for various time points (30–150 min) and analyzed for the appearance of CD107a on the cell surface via flow cytometry. Representative flow plots of unstimulated and stimulated CD8+ NT and E-Syt1 knockdown cells activated for 60 min. Graph of the percentage of CD107a + CD8 + T cells across all activation time points. Plots are a mean ± SEM of at least three independent experiments. Significance was determined by two-way ANOVA with Bonferroni's multiple comparisons test against control NT siRNA (*$P < 0.0332$). (C) Measurement of lymphocyte-mediated cytotoxicity of siRNA-treated human CD8+ T cells against P815 target cells using EarlyTox Caspase-3/7 NucView-488 assay. The percentage of P815 target cell death was tested every 5 min for over 5 h. Plots are a mean ± SEM of at least three independent experiments. Significance was determined by two-way ANOVA with Bonferroni's multiple comparisons test against control NT siRNA (**$P < 0.0021$, ****$P < 0.0001$). (D–F) Flow cytometry analysis of CD3 + CD8+ gated T cells from human PBMCs using signaling events measured by phosphorylation levels of early TCR signaling proteins ZAP70 at Y292 (B) LAT at Y226 (C) and PLC-γ1 at Y783 (D) upon stimulation with anti-CD3 and CD28 antibody-coated beads for indicated times. Plots show the change in median fluorescence intensity (MFI) between the indicated time of stimulation and $t = 0$. Plots are a mean ± SEM of at least three independent experiments. Significance was determined by two-way ANOVA with Bonferroni's multiple comparisons test against control NT siRNA CD8+ T cells (*$P < 0.0332$, **$P < 0.0021$). Source data are available online for this figure.

siRNAs against E-Syt proteins, thus ruling out an off-target effect of CRISPR guide RNAs (Appendix Fig. 2A–D). Collectively, these data suggest that downregulation of mainly E-Syt1 in CD8+ T cells results in a significant enhancement of effector functions such as degranulation and cytotoxicity, whereas the lack of expression of both E-Syt1 & 2 in CD4+, CD8+, and Jurkat T cells increases TCR signaling.

## Loss of E-Syt2 enhances IL-2 production in CD4+ T cells in resting and activated conditions

To further evaluate whether the increased TCR signaling of both early and late signaling molecules in E-Syt-KO cells modulates the key cellular functions of T cells, we tested their ability to produce IL-2. Hence, we analyzed E-Syt KO and WT-Jurkat cells, in resting state and in activated state via anti-CD3/28 coated beads and monitored intracellular cytokine production by flow cytometry. Results showed that under resting conditions, a higher number of IL-2 expressing Jurkat T cells were seen in E-Syt1 KO (approximately fivefold increase), E-Syt2 KO (13-fold increase), and E-Syt1 & 2 DKO (11-fold increase) cells compared with WT cells (Fig. 3A). Interestingly, only E-Syt2 KO cells displayed an ~2.5-fold increase in IL-2 production compared with WT cells upon TCR activation (Fig. 3B). The increased percentage of IL-2 expressing Jurkat cells (Fig. 3A,B), as well as the total amount of IL-2 produced by each cell (Appendix Fig. 3), positively correlated

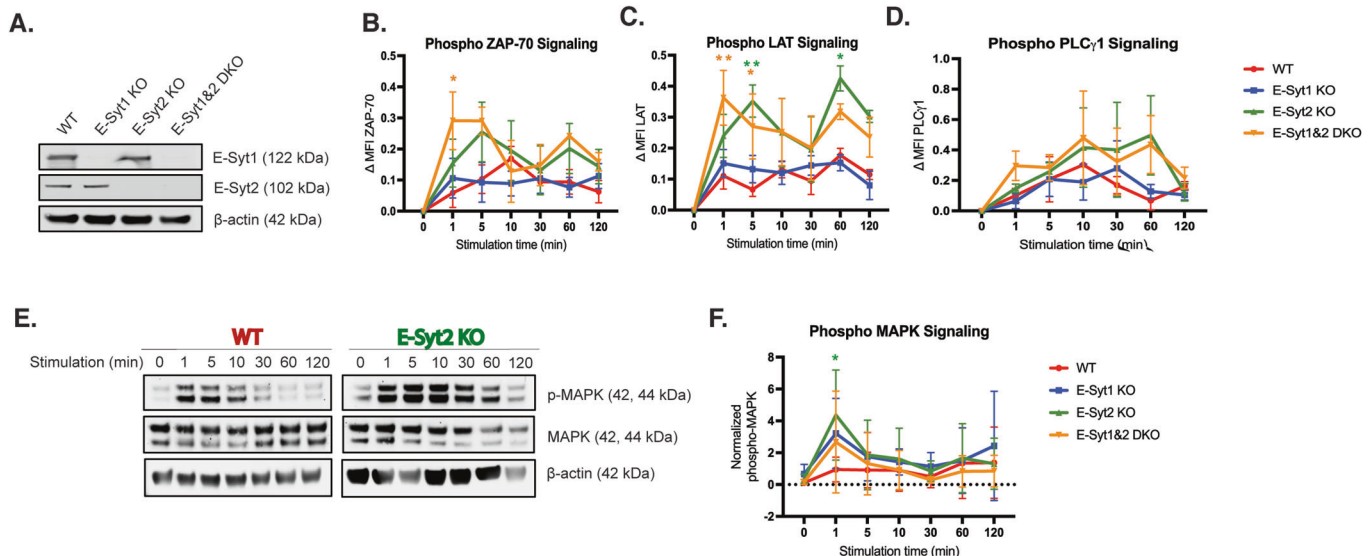

**Figure 2. Absence of E-Syts increases TCR signaling kinetics in Jurkat cells.**

(A) Western blot analysis of the expression of E-Syt proteins in cell lysates from CRISPR–Cas9 knockout single-cell Jurkat T-cell clones E-Syt1 KO, E-Syt2 KO, E-Syt1&2 DKO, and WT-Jurkat cells. (B–D) Flow cytometry analysis of Jurkat cell signaling events measured by phosphorylation levels of early TCR signaling proteins ZAP70 at Y292 (B), LAT at Y226 (C), and PLC-γ1 at Y783 (D) upon stimulation with anti-CD3 and CD28 antibody-coated beads for indicated times. Plots show the change in median fluorescence intensity (MFI) between the indicated time of stimulation and $t = 0$. Plots are a mean ± SEM of at least three independent experiments. Significance was determined by two-way ANOVA with Bonferroni's multiple comparisons test against control WT Jurkat (*$P < 0.0332$, **$P < 0.0021$). (E) Quantification of MAPK phosphorylation. Representative western blot showing levels of p44/42 MAPK phosphorylation at residuesT202/Y204 from cell lysates of WT control and E-Syt2 KO Jurkat cell line stimulated with anti-CD3 and CD28 antibody-coated beads for indicated times. Total MAPK and β-actin were used as loading controls. (F) Densitometry graph showing normalized phosphorylation of MAPK from three independent experiments. Plots are a mean ± SEM of at least three independent experiments. Significance was determined by two-way ANOVA with Bonferroni's multiple comparisons test against control WT Jurkat (*$P < 0.0332$). Source data are available online for this figure.

with the enhanced phosphorylation activity of MAPK (Fig. 2F; Appendix Fig. 2A,D). Of note, all the Jurkat cell lines displayed equivalent capabilities in IL-2 production when stimulated with the strong agonists phorbol myristate acetate (PMA) & Ionomycins as controls (Fig. 3C). Nonetheless, to confirm that the observed phenotypes were attributable to respective E-Syt protein knockouts, we performed rescue experiments in which each E-Syt KO clone was complemented with either the corresponding GFP-E-Syt protein or GFP-LifeAct as a control. For this, we transfected E-Syt KO and WT-Jurkat cells with GFP-E-Syt1, GFP-E-Syt2, or GFP-LifeAct and measured the level of IL-2 production under resting conditions or upon activation. Results from these studies revealed that the number of IL-2-producing cells significantly decreased when E-Syt KO cells were complemented with the corresponding E-Syt protein in both resting and activating conditions (dashed bars) (Fig. 3A,B). To further investigate whether this effect is conserved in primary cells we performed similar experiments in E-Syt knockdown (KD) PBMCs. These data showed an increase in IL-2 levels in the E-Syt2 KD CD4+ T cells and in the E-Syt1&2 DKD CD8+ T cells (Fig. 3D,E) and correlated with those in Jurkat E-Syt-KO cell lines. The enhanced effect on IL-2 production pathways upon E-Syt2 deletion resembles that described when silencing other DAG regulators (Arranz-Nicolas et al, 2020), therefore suggesting a common mechanism leading to enhanced signaling and functionality of T cells. Altogether these data indicate that E-Syt1 and E-Syt2 play an interdependent regulatory role at steady state for proper activation of T cells.

## E-Syt1 and -2 localization correlate with DAG distribution patterns during IS formation

Previous studies have shown that DAG is a critical secondary messenger for TCR signaling, identifying a highly regulated gradient of DAG centered at the IS upon TCR activation (Chauveau et al, 2014; Huse, 2014). In vitro data have shown that the SMP domain of E-Syt proteins transports glycerophospholipids such as DAG down its concentration gradient between the ER and PM (Herdman and Moss, 2016). We investigated the spatial distribution of E-Syt proteins in relation to DAG and T-cell signaling molecules before and after TCR stimulation. As such, we performed Super-Resolution Microscopy in WT-Jurkat cells co-transfected with either mCherry-E-Syt1 or -2 along with a GFP-DAG biosensor consisting of the C1 domain of PKCγ fused to GFP (C1-GFP) (Gawden-Bone et al, 2018; Quann et al, 2009). Transfected Jurkat cells co-expressing mCherry-E-Syt and C1-GFP were seeded onto either resting state (poly-lysine treated only glass surface) or activated conditions (poly-lysine plus anti-CD3 and -CD28 antibodies). Cells were then immunostained for subcellular markers. Imaging was performed at the Z-planes closest to the glass-cell membrane contact area. We found that under resting state, WT-Jurkat cells exhibited a uniform distribution of DAG throughout the PM without specific areas of accumulation (Fig. 4A, arrows) as previously described by others (Chauveau et al, 2014; Gawden-Bone and Griffiths, 2019). Under these conditions, mCherry-E-Syt1 and -2 also showed a diffused localization pattern, mainly on the cell membrane periphery with little to no

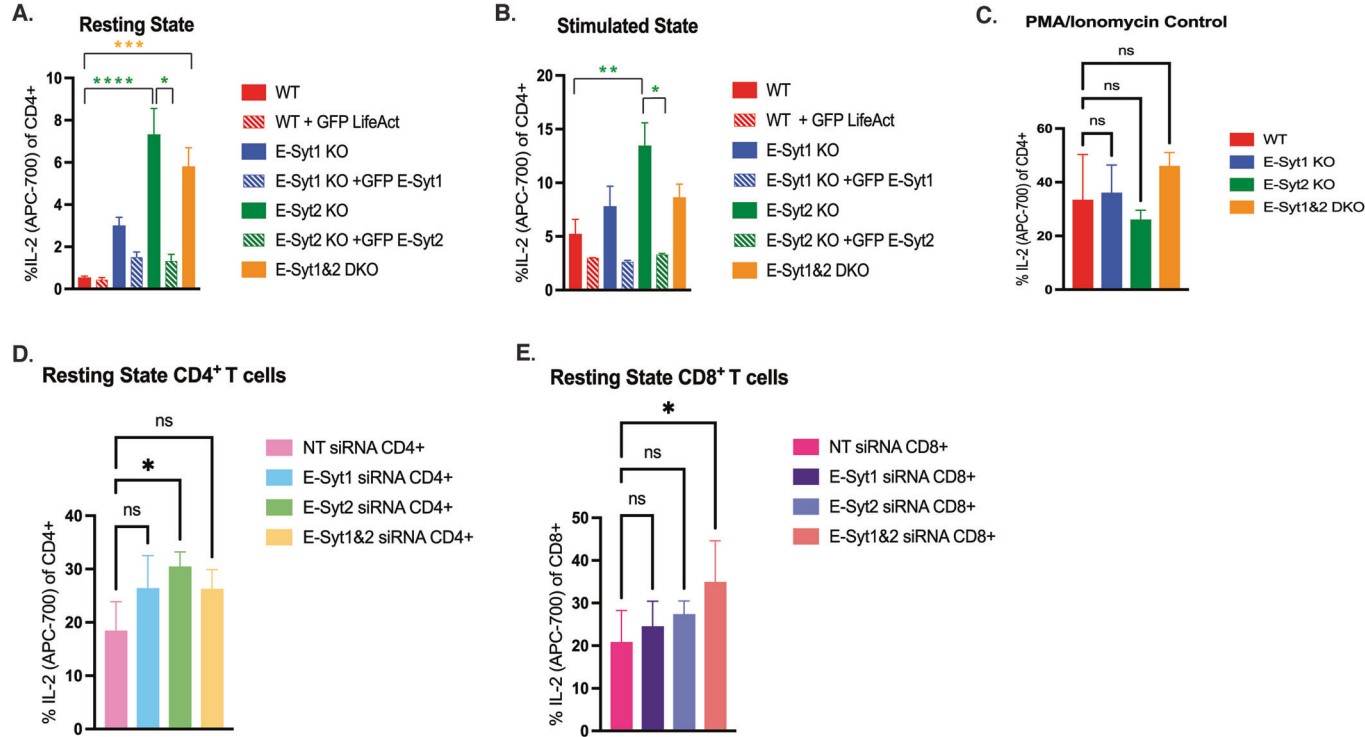

**Figure 3. E-Syt proteins regulate IL-2 production in both Jurkat and primary T cells.**

(**A**) Intracellular staining of IL-2 measured by flow cytometry in WT and E-Syt KO Jurkat cell lines at resting state (solid bars). Rescue experiments of unstimulated WT, E-Syt1, or E-Syt2 KO Jurkat cell lines transfected with either GFP-LifeAct or GFP-E-Syt1/E-Syt2, respectively, are represented with dashed-line bars. GFP+ cells were sorted and used for the following experiments for maximum accuracy. The data shown represent the mean ± SEM of at least five independent experiments. Significance was determined by one-way ANOVA with Bonferroni's multiple comparisons test against control WT Jurkat (*$P < 0.0332$, ***$P < 0.0002$, ****$P < 0.0001$). (**B**) Intracellular staining of IL-2 in WT and E-Syt KO Jurkat cell lines stimulated with anti-CD3 and -CD28 coated beads for 12 h (solid bars). Rescue experiment of stimulated WT, E-Syt1, or E-Syt2 KO Jurkat cell lines transfected with either GFP-LifeAct or GFP-E-Syt1/E-Syt2, respectively, are represented with dashed-line bars. GFP+ cells were sorted and used for the following experiments for maximum accuracy. The data shown represent the mean ± SEM of at least five independent experiments. Significance was determined by one-way ANOVA with Bonferroni's multiple comparisons test against control WT Jurkat (*$P < 0.0332$, **$P < 0.0021$). (**C**) Intracellular staining of IL-2 in WT and E-Syt KO Jurkat cell lines stimulated with PMA and Ionomycin as controls for 12 h. The data shown represent the mean ± SEM of at least five independent experiments. Significance was determined by one-way ANOVA with Bonferroni's multiple comparisons test against control WT Jurkat. (**D, E**) Intracellular staining of IL-2 in CD3 + CD4+ (**D**) and CD3 + CD8+ (**E**) T cells from PBMCs were stimulated with anti-CD3 and -CD28 coated beads for 12 h. The data shown represent the mean ± SEM of four independent experiments from same donor cells. Significance was determined by one-way ANOVA with Bonferroni's multiple comparisons test against control WT Jurkat (*$P < 0.0332$). Source data are available online for this figure.

accumulation in the center of the cell, resembling DAG and CD4 localization. In contrast, activated Jurkat cells displayed a strong patch of DAG accumulation at the PM, as previously described (Fig. 4A, arrowheads) (Gawden-Bone et al, 2018). These DAG clusters were highly enriched with the TCR co-receptor CD4 (Fig. 4A). Interestingly, under these conditions, mCherry-E-Syt1 mainly localized as a ring-like pattern overlapping and surrounding the clusters of DAG and CD4 (Fig. 4A). In contrast, mCherry-E-Syt2 showed a punctate pattern concentrated in DAG-rich regions. Colocalization analysis demonstrated that both mCherry-E-Syt1 and -2 displayed a higher Pearson's correlation coefficient (PCC) with DAG in resting and activated conditions than with Early Endosomal Antigen 1 (EEA1); a marker that cycles between the PM and early endosomes (Fig. EV2). E-Syt1 and E-Syt2 subcellular localization was also compared to different organelle markers like KDEL receptor (ER), CellMask (PM), and DRAQ5 (nucleus). Images showed that both E-Syt1 and E-Syt2 distribution resembles mainly the PM and to a lesser extent the ER staining (Appendix Fig. S4). These results demonstrate that both E-Syt1 and -2 change

their localization patterns upon TCR activation and spatiotemporally colocalize with DAG-rich regions, therefore locating E-Syt proteins as potential modulators of DAG levels in the PM.

## Loss of E-Syt proteins results in the accumulation of DAG and signaling proteins at the PM in the resting state

To better understand the mechanism by which loss of E-Syt proteins increases TCR signaling and functionality, we tested our hypothesis that the absence of E-Syt proteins interferes with the distribution pattern of DAG at the PM. To this end, we evaluated the steady-state localization of DAG in E-Syt1 KO, E-Syt2 KO, and E-Syt1&2 DKO Jurkat cells under resting state and activated conditions. Strikingly, we found that in the resting state, all E-Syt KO cell lines displayed a large patch of DAG accumulation at the PM in addition to the homogenous staining of the remainder of the cell membrane comparable to that of Jurkat WT cells in the resting state (Fig. 4B, arrowheads). This contrast in the pattern of DAG highly resembled the patched distribution of DAG in the activated

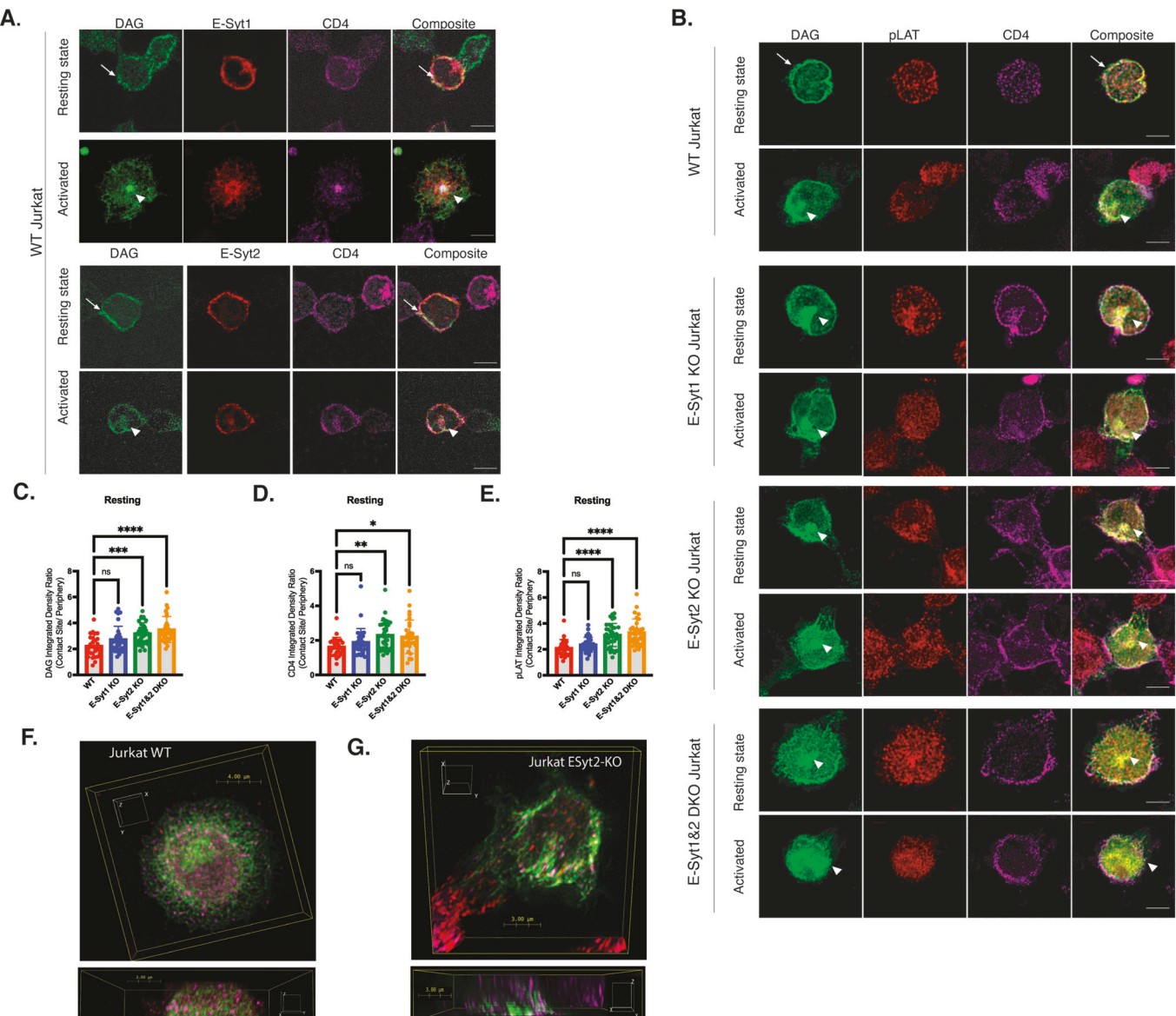

**Figure 4. E-Syt proteins modulate DAG and signaling markers in resting-state Jurkat cells.**

(A) Representative Super-Resolution confocal images of WT-Jurkat cells transfected with GFP-PKCg-C1 probe (DAG), mCherry-E-Syt1/E-Syt2, and stained with anti-CD4 antibody at resting state (attached to poly-lysine-treated glass coverslips) or activated state (attached to poly-lysine plus anti-CD3 and -CD28 antibodies treated with glass coverslips). Arrowheads highlight DAG accumulation at the PM. Arrows indicate uniform DAG distribution throughout the PM. Scale bars: 5 μm. (B) WT, E-Syt1 KO, E-Syt2 KO, and E-Syt1&2 DKO Jurkat cells transfected with GFP-PKCg-C1 probe (DAG) and stained with anti-CD4 and anti-pLAT antibodies as synaptic markers in resting and activated states. Arrowheads highlight DAG accumulation at the PM. Arrows indicate uniform DAG distribution throughout the PM. Scale bars: 5 μm. (C–E) Quantification of the ratio of the integrated fluorescent density of DAG (C), CD4 (D), and p-LAT (E) between the contact site area (DAG-rich area) and the cell periphery (low DAG signal). Data are shown as mean ± SEM of at least three independent experiments (WT $n = 24$ cells; E-Syt1 KO $n = 31$ cells; E-Syt2 KO $n = 34$ cells; E-Syt1&2 KO $n = 30$ cells). Significance was determined by one-way ANOVA with Bonferroni's multiple comparisons test against control WT Jurkat (*$P < 0.0332$, **$P < 0.0021$, ***$P < 0.0002$, ****$P < 0.0001$). (F, G) 3D reconstruction of Z-stacks from resting-state WT (F) and E-Syt1&2 DKO (G) Jurkat cells transfected with GFP-DAG (green) and stained for p-LAT (red) and CD4 (magenta). Source data are available online for this figure.

WT cells (Fig. 4B, arrowheads). E-Syt2 and E-Syt1&2 DKO cells displayed a stronger DAG signal than the control and E-Syt1 KO cells in resting conditions as shown by the images and quantification of the fluorescent integrated density ratio of DAG (Fig. 4B,C). To further examine whether the altered localization of DAG in the resting state of E-Syt KO cells is consistent with changes in the amount of T-cell signaling proteins, we also measured the integrated density ratios of CD4 and p-LAT and found these signals to be upregulated in E-Syt2 and E-Syt1&2 DKO cells (Fig. 4D,E). Colocalization analysis of DAG and CD4 revealed a marked increase in PCC of GFP-DAG with CD4 in all E-Syt KO cells in the resting state. However, there were no significant differences in the colocalization of GFP-DAG and CD4 between E-Syt KO and WT cell lines under activated conditions (Appendix

Fig. S5). Under these conditions, we also observed an increase in both total and phosphorylated ZAP70 as measured by the ratio of integrated density values between the center and cell periphery (Fig. EV3). Even so, activated E-Syt KO cells showed slightly increased accumulation of DAG compared to activated WT cells, usually presenting as a larger patch and small clusters scattered throughout the membrane. These findings strongly indicate that E-Syt proteins influence DAG distribution in the PM a seen in the 3-dimensional (3D) reconstruction images (Fig. 4F,G). Moreover, we found that the lack of their expression induces the accumulation of DAG in patch-like structures in the resting state which concentrates upstream signaling proteins (CD4, LAT, and ZAP70) that mimic the hallmarks of activated T cells. Conversely, the analysis of DAG distribution in cells overexpressing E-Syt1 did not show any statistical difference (Appendix Fig. S6).

To analyze if the DAG accumulation in PBMCs with down-regulated E-Syt proteins leads an increased number of polarized lytic granules that could account for the enhanced cytotoxicity and degranulation previously seen, we measured the colocalization of DAG and CD63, a lytic granule marker (Blott and Griffiths, 2002; Stinchcombe et al, 2006) (Appendix Fig. S7). Data showed that E-Syt knockdown in PBMCs yields an increased colocalization of CD63 with DAG. Collectively, this indicates that the absence of E-Syt proteins induces the accumulation of DAG and other signaling proteins on the surface of T cells, which in turns promotes lytic granule polarization and subsequent effector functions.

## E-Syt2 temporally controls DAG distribution at the PM in resting conditions

Given our previous findings that lack of expression of E-Syt2 in conjunction with E-Syt1 induces DAG accumulation at the PM, enhances T-cell signaling events, and critical effector functions, we sought to determine if the kinetics and localization of DAG distribution at the PM of T cells were affected by the loss of either E-Syt protein. To directly investigate the role of E-Syt proteins in modulating local concentrations of DAG at the PM in live cells, we performed total internal reflection fluorescence (TIRF) microscopy with our Jurkat cell lines expressing a biosensor containing a high-affinity C1ab domain from PRKD1 fused to GFP (Zewe et al, 2020). The combination of this modified biosensor (GFP-C1ab) has been reported to have greater specificity for DAG at the PM along with TIRF microscopy, which only allows for the detection of fluorescent molecules within the depth of the evanescent field (100–150 nm), provided more accurate conditions for analyzing DAG dynamics at the PM. E-Syt KO and WT cell lines expressing GFP-C1ab were deposited onto non-activating surfaces and imaged using TIRF microscopy over the course of 5 min. Total integrated density analysis of DAG signals in TIRF movies indicated that in resting-state Jurkat cells the absence of E-Syt2 and E-Syt1&2 DKO cells increases DAG signals on the PM compared to WT cells (Fig. 5A,B). Analysis of the ratio of integrated density values (value at the area of greatest DAG signal divided by value at a similar size area of the periphery of the cell) revealed a spike increase within the first minute of E-Syt2 KO cells interacting with the glass surface (Fig. 5B). These data correlate with our previous results in which resting-state fixed E-Syt KO cells showed larger clusters of DAG at the PM, which concentrated signaling proteins (Figs. 4 and EV3;

Appendix Fig. S5) and thus ruled out the possibility of lipid clustering artifacts due to the fixation process.

## E-Syt1&2 together modulate DAG levels at the PM upon stimulation

To further investigate whether DAG dynamics are also altered in E-Syt KO cells upon stimulation we performed experiments like those described above. However, here, cell lines were deposited onto stimulatory surfaces containing anti-CD3 and CD28 antibodies. Unlike our previous results in resting-state conditions which showed that only E-Syt2 KO cells displayed a spike in DAG accumulation within the first few minutes of contact with the surface, we observed a steadily enhanced level of DAG accumulation throughout the span of 5 min in both E-Syt1&2 DKO and E-Syt2 KO cells (Fig. 5C). Upon analysis of single images from the TIRF movies, we observed an overall greater DAG signal throughout the cell in DKO than in WT-Jurkat T cells (Fig. 5D). Overall, these results demonstrate that in the resting state, E-Syt1 and E-Syt2 control basal levels of DAG at the PM and their absence induces the accumulation and clustering of DAG, which is sufficient to trigger phosphorylation of both early and late signaling proteins and enhance the production of IL-2 (Figs. 2 and 3).

## DAG production at the PM of T cells requires PLCγ activity

To analyze the source of DAG accumulation at the PM, we exploited the use of two key inhibitors that regulate the production (phospholipase C inhibitor, U73122) and consumption (diacylglycerol kinase inhibitor, R59022) of DAG. TIRF experiments were performed in similar conditions as previously described in the presence and absence of each inhibitor. The data showed that inhibition of PLCγ, regardless of the activation state, reduces the amount of DAG present in the PM (Fig. EV4). On the contrary, treatment with DGK inhibitors did not produce significant effects in resting conditions, but significantly enhanced DAG levels upon activation in the E-Syt2 KO Jurkat cells (Fig. EV4). To correlate the effect of inhibitor treatments with downstream functions, we measured IL-2 production. As predicted, we observed decreased levels of IL-2 in PLCγ inhibitor-treated cells regardless the activation state. However, treatment with the DGK inhibitor slightly decreased IL-2 production in all cell lines except E-Syt2 KO Jurkat cells (Fig. EV5) This interesting observation requires further investigation to dissect their mechanism of action. Altogether, our data suggest that the source of DAG accumulated in our system depends upon PLCγ activity and that T cells require both DGKs and E-Syt protein activity to modulate DAG levels at the PM.

## Discussion

In this study, we identified a novel regulatory mechanism of DAG-mediated signaling for T-cell effector functions based on inter-organelle lipid transport proteins that modulate lipid homeostasis at the PM. The way by which lipid transfer occurs at the PM-ER contact sites in T cells remains elusive. Herein, we uncovered a new role of E-Syt1 and -2 in modulating DAG dynamics at the PM in

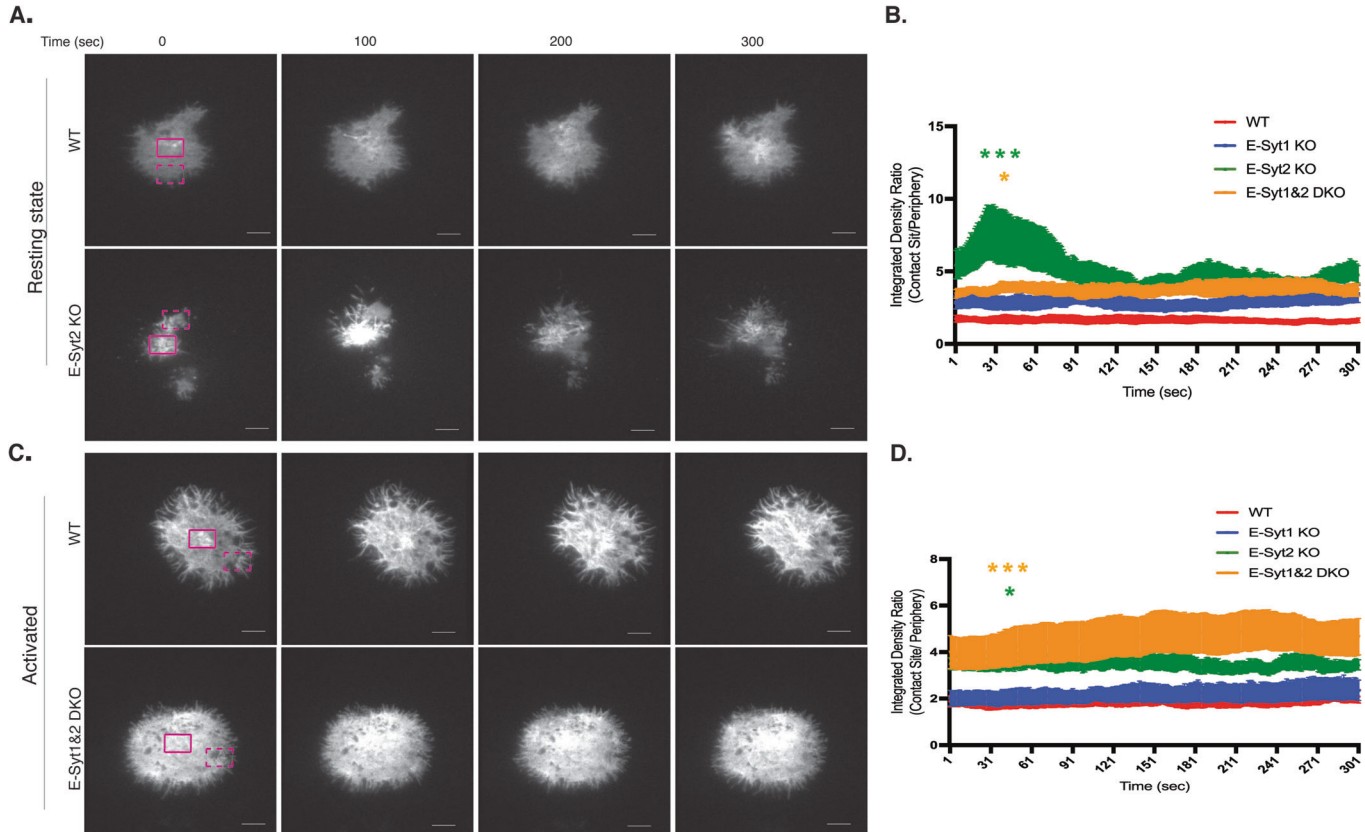

**Figure 5. E-Syt2 modulates DAG levels at the PM.**

(**A**) Representative images at 0, 100, 200, and 300 s from TIRF movies of WT control or E-Syt KO Jurkat cells transfected with the C1ab domain of PRKD1 fused to GFP construct for PM DAG detection (GFP-C1ab) deposited onto poly-lysine coated glass chamber (resting state, **A**). A region of interest for the contact site (DAG-rich area) and cell periphery (low DAG signal) are marked in solid magenta and dashed magenta respectively. Scale bars: 3 μm. (**B**) Quantification of the ratio of the integrated fluorescent density of GFP-C1ab-Pkrd1 (DAG) between the contact site area (DAG-rich area) and the cell periphery (low DAG signal) in resting-state Jurkat cells. Data are shown as mean ± SEM of at least three independent experiments (WT $n = 8$ cells; E-Syt1 KO $n = 8$ cells; E-Syt2 KO $n = 8$ cells, E-Syt1&2 KO $n = 8$ cells). Significance was determined by two-way ANOVA with Bonferroni's multiple comparisons test against control WT Jurkat (*$P < 0.0332$, ***$P < 0.0002$). (**C**) Representative images at 0, 100, 200, and 300 s from TIRF movies of WT control or E-Syt KO Jurkat cells transfected with GFP-C1ab deposited onto poly-lysine coated glass chambers coated with anti-CD3 and anti-CD28 (activated state, **C**). A region of interest for the contact site (DAG-rich area) and cell periphery (low DAG signal) are marked in solid magenta and dashed magenta, respectively. Scale bars: 3 μm. (**D**) Quantification of the ratio of the integrated fluorescent density of GFP-C1ab-Pkrd1 (DAG) between the contact site area (DAG-rich area) and the cell periphery (low DAG signal) in activated state Jurkat cells. Data are shown as mean ± SEM of at least three independent experiments (WT $n = 8$ cells; E-Syt1 KO $n = 8$ cells; E-Syt2 KO $n = 8$ cells, E-Syt1&2 KO $n = 8$ cells). Significance was determined by two-way ANOVA with Bonferroni's multiple comparisons test against control WT Jurkat (*$P < 0.0332$, ***$P < 0.0002$). Source data are available online for this figure.

T cells. E-Syts have been studied in various cell types; however, their precise role in vivo physiological processes remains poorly understood given the viability, fertility, and grossly normal development of mutant mice lacking all three E-Syts in the absence of a challenge to the immune system (Sclip et al, 2016; Tremblay and Moss, 2016). The lack of an evident phenotype in these mutant mice could be due in part, to the largely redundant nature of inter-organelle membrane contact site proteins that act in lipid transport. An alternative explanation may rest in the cell-specific roles of these proteins, as demonstrated here with primary CD8[+], CD4[+] T cells and what other groups have shown in pancreatic β cells with the regulation of insulin secretion by E-Syt1 (Beichen Xie, 2019). Their different tissue distribution and subcellular localization suggest that they may have distinct functions in different cell types. Nonetheless, the strong evolutionary conservation of E-Syts from plants to yeast to animals strongly suggests an important role

in cellular functions (Bian Saheki and De Camilli, 2018); Herdman and Moss, 2016; Tremblay and Moss, 2016).

Using a combination of immunological and imaging approaches, we showed that E-Syt1 and -2 are strategically positioned on the synaptic membrane interphase controlling DAG dynamics and fine-tuning T-cell signaling kinetics, cytokine production, and even cytotoxic functions (Fig. 6A,B). We suspect that this localization acts as a non-vesicular transport mechanism for lipid regulation during the activation of T cells. In support of this, we observed greater phosphorylation of MAPK and IL-2 production in cells lacking E-Syt2 but not E-Syt1 only, both of which are downstream of DAG. In addition, under resting conditions, the observed patches of DAG in our E-Syt KO cells co-localized with clusters of CD4 and signaling proteins, which in turn, may induce phosphorylation of upstream (LAT, ZAP70) and downstream (MAPK) molecules. This suggests it is possible that

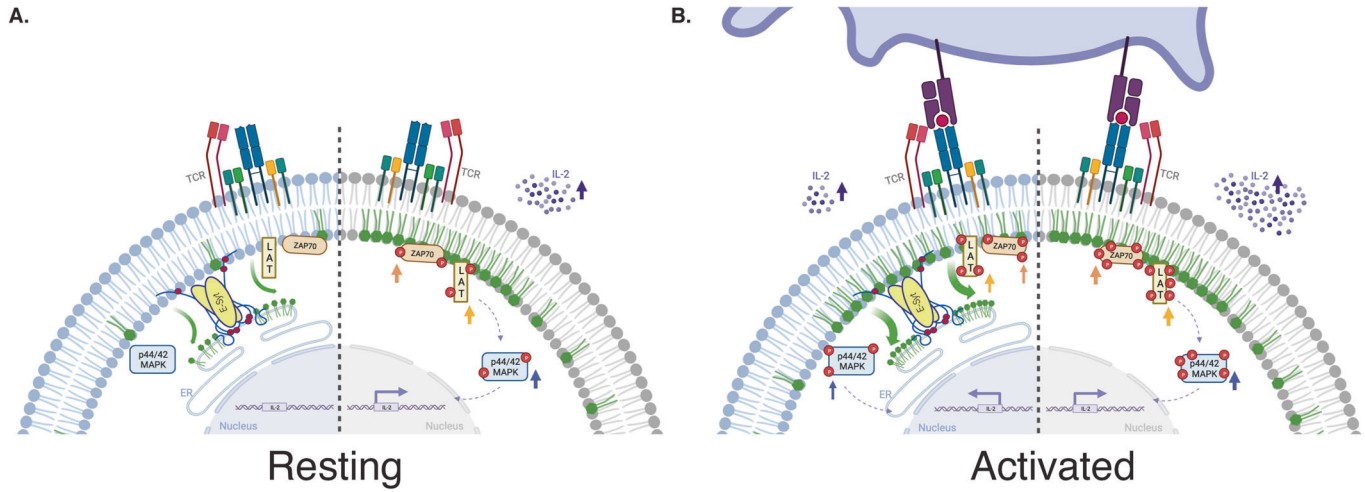

**Figure 6. E-Syt proteins modulate signaling and functionality of T cells via PM DAG levels.**

(A) Left: E-Syt dimer located between the ER and PM in the unstimulated T cell. Green arrows indicate the transfer of DAG (green lipid structure) from the PM to the ER. No phosphorylation of TCR signaling proteins (ZAP70, LAT, and MAPK) or IL-2 production. Right: The absence of E-Syt protein dimer at ER-PM junction sites leads to the accumulation of DAG (green lipid structure) at the PM. DAG accumulation at the PM leads to the phosphorylation of TCR signaling proteins (ZAP70, LAT, and MAPK) and eventually IL-2 production. (B) Left: E-Syt dimer located between the ER and PM in the activated T-cell mediate the transfer of DAG (green lipid structure) from the PM to the ER. Normal phosphorylation of TCR signaling proteins (ZAP70, LAT, and MAPK) and IL-2 production expected. Right: The absence of E-Syt protein dimer at ER-PM junction sites in an activated T cell leads to the vast increase of DAG (green lipid structure) at the PM. DAG accumulation leads to the enhanced phosphorylation of TCR signaling proteins (ZAP70, LAT, and MAPK) and IL-2 production. This figure was created with BioRender.com. Source data are available online for this figure.

E-Syt2 primarily conducts the steady-state non-vesicular lipid transfer at DAG-rich sites on the PM.

We identified two cellular states in which E-Syt1 and -2 distinctly modulated T-cell activation. In accordance with previous findings, E-Syt2 permanently localizes at ER-PM junction sites, independent of calcium flux, and acts as a steady-state transporter of DAG to the ER (Saheki et al, 2016). Such localization parallels our results of enhanced phosphorylation of signaling proteins, IL-2 production, and colocalization of DAG and CD4 in E-Syt2 KO cells under resting and activated conditions. Conversely, E-Syt1 alone, which is mainly located at the ER membrane and requires high intracellular $Ca^{2+}$ levels for tethering to the PM (Idevall-Hagren and De Camilli, 2015) does not modulate DAG levels at the PM under resting conditions according to our results. We observed in our activated Jurkat cell model that the downregulation of E-Syt1 alone was insufficient to have a significant impact on T-cell signaling kinetics, functionality, or cytokine production. We believe this may be due to an alternative compensatory mechanism, namely, DGKs that enzymatically control DAG homeostasis in activated T cells. Previous studies proposed that during synaptic formation, DGKα limits the diffusion of DAG at the IS periphery (Chauveau et al, 2014), while DGKζ controls PM DAG levels for signaling pathways (Topham and Epand, 2009). Both isoforms require prior TCR engagement (Krishna and Zhong, 2013). DGKα requires phosphoinositide 3-kinase (PI3K) activity for proper peripheral localization (Chauveau et al, 2014), while DGKζ activity involves PKC-dependent phosphorylation of its MARCKS domain (Gharbi et al, 2011). Surprisingly, we observed a significant increase in cellular degranulation and cytotoxicity in activated primary CD8[+] T cells with downregulated E-Syt1 expression. A possible explanation is cell-type-specific differences in the subcellular organization between CD4[+] and CD8[+] T cells. Our data suggest

that in the resting state, E-Syt2 primarily regulates basal DAG production at the membrane, while both E-Syt1 and -2 are involved in downmodulating DAG levels in the activated state.

Altogether, our data strongly suggest that E-Syt1 and -2 downmodulate DAG-mediated signaling in T cells, and to our knowledge, we are the first to characterize their crucial localization at DAG-rich regions in this cell type (Fig. 6A). These findings further highlight their implication as ideal targets for effector functions of T cells. We observed that in the absence of E-Syts, cells bypass TCR activation and are in a pre-activated state, independent of standard stimulation (Fig. 6B). This unique state highlights unexplored processes of T-cell functioning, such as the effect on cell migration and even the IS formation. Further imaging studies for actin polarization and localization of phospho-signaling proteins are necessary.

Limitations of this study include restrictions in our live-cell imaging methods, which only allow us to image DAG at the PM but not its transfer to the ER. Additionally, we were limited in the available DAG biosensors as visualization tools. Thus, we used two different sensors to ensure that the observed DAG occurrence we saw was indeed at the PM. Alternatively, we also considered a potential effect of the lack of E-Syt proteins leading to an induction of apoptosis after TCR stimulation. However, this possibility was discarded since we did not observe an increase in apoptosis markers upon activation (Appendix Fig. S8).

In summary, the present work characterizes the distinct role of E-Syt1 and -2 in modulating DAG levels at the PM of human T cells. Though DGKs terminate DAG signaling in T cells via enzymatic activity, our findings uncover an additional regulatory mechanism of non-vesicular DAG transfer via E-Syt1 and -2. These two separate mechanisms of DAG regulation mediate proper signaling and avoid hyper-activation of T cells upon TCR

engagement. Moreover, the current study elucidates the underlying molecular mechanism of DAG regulation by identifying the colocalization of E-Syt1 and -2 with DAG-rich clusters during IS formation. By extension, our findings suggest that the inactivation of E-Syt proteins represents a potential target for improved T-cell responses.

# Methods

## Reagents and antibodies

Mouse anti-CD3 (OKT3 functional grade purified) was purchased from BD Pharmingen (cat #:16-0037-85, San Jose, CA, USA). Mouse anti-Munc13-4 was purchased from Santa Cruz Biotechnology (cat #: sc50465, Dallas, TX, USA). Mouse anti-F-actin was purchased from Sigma (cat #: A-5441, St. Louis, MO, USA). Mouse anti-green fluorescent protein (GFP) was purchased from Roche (cat #: 11814460001, Indianapolis, IN). Rabbit anti-ESYT1 was purchased from NOVUS Biologicals (cat #: NBP1-84741, Centennial, CO, USA). Rabbit anti-ESYT2 was purchased from ATLAS Antibodies (cat #: HPA002132, Sweden). Rabbit anti-E-SYT3 was purchased from NOVUS Biologicals (cat #: NBP1-91354, Centennial, CO, USA). Mouse anti-tubulin was purchased from Santa Cruz Biotechnology (cat #: sc-32293, Dallas, TX, USA). Rabbit CD63 was purchased from Abcam (cat #: ab134045, Cambridge, UK). Mouse anti-EEA1 was purchased from BD Transductions (cat #: 610457, Franklin Lakes, NJ, USA). Rabbit anti-LAT was purchased from Cell Signaling (cat #: 45533S, Danvers, MA, USA). Mouse anti-CD4 was purchased from Biolegend (cat #: 300502, San Diego, CA, USA). Rabbit anti-phospho p44/42 MAPK (T202/Y204) was purchased from Cell Signaling (cat #: 9101S, Danver, MA, USA). Rabbit anti-MAPK 1/2 was purchased from EMD Millipore Sigma (cat #: 50095, Munich, Germany). Secondary antibodies donkey anti-rabbit-IRDye800 CW-conjugated (cat #: 926-32213) and donkey anti-mouse-IRDye680-RD conjugated (cat #: 926-68072) were purchased from LiCor (Lincoln, NB, USA).

## Cell lines, culture, and stimulation

Human leukemic Jurkat T cells were obtained from the American Type Culture Collection (clone E6-1). Jurkat T cells were cultured in complete RPMI-1640 medium supplemented with 10% FBS and 5% Penn-Strep. Murine P815 mastocytoma cell line was used from the American Type Culture Collection (Manassas, USA) and cultured in complete DMEM medium supplemented with 10% FBS and 5% Penn-Strep. Primary CD8$^+$ T cells were obtained from unidentified human samples from the Human Immunology Core at the University of Pennsylvania. Isolated human CD8$^+$ cells were expanded using Dynabeads (Human T-Expander CD3/CD28 from Life Technologies) and human recombinant IL-2 for 5 days in complete medium. After, beads were removed using a magnet and cells were used for various experiments.

CRISPR–Cas9 knockout Jurkat T cells were generated by electroporating single guide RNA (sgRNA) specifically designed to target genomic DNA together with Alt-R S.p Hifi Cas9 nuclease V3 (Integrated DNA Technologies). The sgRNA sequences used were the following: ESYT1 gRNA: CGGTGCTGACTTCATTCGGG and ESYT2 gRNA: GTGATCCTGGAACCGTTGAT. Seventy-two hours post-transfection, serial dilutions were made for single-cell cultures into 96-well plates. Single-cell clones were screened using western blotting with actin as a loading control. Successful E-Syt1, E-Syt2, and E-Syt1&2 knockout clones were expanded, and stocks were frozen.

## Plasmids and transfection

Jurkat T cells and Human primary CD8$^+$ T cells were electroporated using the Neon Transfection System (ThermoFisher; at 1350 V; 10 ms pulse width; 3 pulses). For transient Munc13-4, E-Syt1, and E-Syt2 silencing, small-interference RNA (siRNA) knockdown experiments were performed. The following siRNA sequences were used: E-Syt1 siRNA: GUACUACAGUGAA-GAACGA E-Syt2 siRNA:GGACAGGACUGACGAAUCU and Munc13-4 siRNA: GACAAGAUCUUCCACAAUA. For a transfection control, nontargeting siRNA sequences were used: 5′-uagcgacuaaacacaucaa-3′; 5′-uaaggcuaugaagagauac-3′; 5′-auguauuggccuguauuag-3′; 5′-augaacgugaauugcucaa-3′ (s#8959). After nucleofection, cells were cultured in RPMI medium without antibiotics and incubated for 48 h before analysis by different experiments. For rescue experiments, we electroporated EGFP-E-Syt1, EGFP-E-Syt2, and EGFP-LifeAct plasmids into the respective E-Syt KO Jurkat cells and WT-Jurkat cells for control.

pCMV-EGFP-PKCg-C1 was used for DAG detection (Addgene plasmid #112266). pmCherry-C1ab-Prkd1 was a gift from Gerry Hammond and Tamas Balla (Addgene plasmid # 139314; http://n2t.net/addgene:139314; RRID:Addgene_139314). mCherry-Lifeact and EGFP-Lifeact were also used for actin controls. mCherry-E-Syt1 and mCherry-E-Syt2 were generated in the lab. mCherry-E-Syt1 was generated by using the SacI and SalI restriction sites on EGFP-E-Syt1 followed by ligation to the mCherry-C1 plasmid. mCherry-E-Syt2 was generated by PCR amplification of pCR4-TOPO-E-Syt2 using BspEI an SalI primers.

## Inhibitor treatments

U73122 was the phospholipase C inhibitor used (TOCRIS). R59022 was the diacylglycerol kinase inhibitor I used (Cayman Chemical). Cells were treated overnight with inhibitors prior to conducting experiments.

## Immunoblot analysis

Cell pellets were lysed [NuPAGE LDS Sample Buffer 1×, pH 8.4] and boiled at 100 °C for 10 min at 1500 rpm on the Eppendorf ThermoMixer. A constant amount of protein per sample was run in sodium dodecyl sulfate-polyacrylamide gel electrophoresis (SDS-PAGE). Odyssey CLx Imaging System (LI-COR) was used to visualize fluorescent signals and Image Studio Lite Software was used for quantification. For phospho-protein analysis by immunoblotting, Jurkat T cells were stimulated with anti-CD3 and -CD28 coated Dynabeads for 1, 5, 10, 30, 60, and 120 min and then lysed as described above including phosphatase inhibitor cocktail (Cell Signaling Technologies). The phosphorylation state of signaling proteins was analyzed by western blot, and normalized values were calculated (phosphor-protein/total protein).

For the co-immunoprecipitation experiments, GFP-STX11 was electroporated into human CD8$^+$ T cells as described above.

Twenty-four hours later, T cells were activated with anti-CD3 and anti-CD28-coated beads then lysed. Co-immunoprecipitation (co-IP) experiments were carried out using co-IP kit as previously published (Thermo Scientific) (Spessott et al, 2017a). Proteins bound to anti-GFP-coated beads were analyzed by SDS/PAGE. Interacting proteins were identified by mass spectrometry.

## Flow cytometry and intracellular cytokine staining

For intracellular cytokine staining, cells were activated for overnight with human Dynabeads (ThermoFisher), anti-CD3/ anti-CD28 coated wells (Invitrogen and BD Pharmingen), or PMA (Sigma) and ionomycin (Sigma). After the first 3 h of stimulation with beads, cells were treated with BrefeldinA (ThermoFisher) for the remainder of the stimulation period (total 12 h). Surface antibodies were first stained using anti-CD4-BV510, anti-CD3-PerCP, and anti-CD45-PE. Cells were then permeabilized and fixed using Cytofix/Cytoperm kit (BD Biosciences). Finally, intracellular proteins were stained with fluorescent antibodies: anti-IL-2-APC700. Data were collected using a Cyto-Flex-S cytometer (Beckman). For quantification, CD3 + CD4+ were gated and cytokine expression levels were assessed. For phospho-protein analysis via flow cytometry, cells were stimulated with anti-CD3 and -CD28 coated Dynabeads for 1, 5, 10, 30, 60, 120 min. After each time point, cells were stained for surface markers using previously described antibodies followed by fixation and permeabilization. Cells were then stained for intracellular signaling proteins: anti-ZAP70 phospho (Tyr292) Alexa Flour488, anti-LAT phospho (Tyr171) Alexa Flour 647, and anti- PLCγ1 phospho (Tyr783) PE (Biolegend). For rescue experiments, WT, E-Syt1 KO, and E-Syt2 KO Jurkat cells were electroporated with GFP-LifeAct, GFP-E-Syt1, and GFP-E-Syt2, respectively. GFP+ cells were sorted using the BD FACS Melody streamline cell sorter (BD Biosciences). Sorted cells were cultured and treated for intracellular cytokine staining as described above.

## Apoptosis assay

Cells were treated with 1 μM Staurosporine (Sigma S6942) for 6 h at 37 °C. Activated condition cells were then stimulated with human Dynabeads (ThermoFisher) for 3 h at 37 °C. After stimulation, cells were washed and incubated with Annexin V Conjugate, CD3-PerCP (clone SK7), and CD4-BV510 (clone OKT4) both of which were from BD Biosciences (San Jose, CA) all in annexin-binding buffer for 15 min at room temperature. Cells were then washed with annexin-binding buffer. Data were collected using a Cyto-Flex-S cytometer (Beckman).

## Cytotoxicity assay

Effector functions such as cell-mediated cytotoxicity of CD8 T cells were evaluated using the EarlyTox Caspase-3/7 NucView-488 Assay Kit (Molecular Devices). CD8+ T cells were activated with anti-CD3 (OKT3) antibody for 15 min. P815 target cells were incubated with membrane-permeable NucView-488 caspase-3/ 7 substrate for 30 min. Effector and target cells were combined in a 10:1 ratio in a black flat-bottom 96-well plate. Separate rows of target and effector cells only were also plated. Fluorescence was measured over 300 min with an interval of 5 min in a fluorescence

microplate reader (ImageXpress® Pico Automated Cell Imaging System-Molecular Devices). The percentage of CD8+ T-cell-mediated cytotoxicity was measured as a percentage of NucView-488 positive cells over a time course.

## Degranulation assay

To assess CD8+ T-cell degranulation, purified CD8+ T cells were incubated in the presence or absence of P815 target cells at a 1:1 ratio for 30, 60, 90, 120, and 180 min at 37 °C. Cells were then stained using the following fluorescently labeled antibodies: anti-CD107a-PE (clone H4A3), CD56-APC (clone NCAM16.2), CD8-FITC (clone SK1), and CD3-PerCP (clone SK7), all of which were from BD Biosciences (San Jose, CA). Data were collected using a Cyto-Flex-S cytometer (Beckman). CD3+CD8+ cells were gated and assessed for surface expression of CD107a.

## Immunostaining

For immunofluorescence labeling, cells were fixed with 4% PFA (10 min RT). After washing with PBS, cells were permeabilized (0.1% Triton in PBS) and blocked (3% BSA) for 30 min RT. BSA was also used for the staining procedure. Cells were incubated with primary antibodies (overnight, 4 °C), then washed with PBS, and incubated with the corresponding secondary Ab (2 h, RT). All coverslips were prepared with poly-L-lysine for adhesion, and for activated conditions, coverslips were treated with human anti-CD3 (BD Pharmingen) and human anti-CD28 (BD Pharmingen). Coverslips were washed three times in PBS and mounted on glass slides using ProLong Gold antifade reagent (Invitrogen).

## Microscopy and image processing

Images were collected on a Leica SP8-STED-3X microscope (Leica Microsystems) using the following settings: GFP 488 (493–549), mCherry 554 (559-644), and Cy5 649 (654-779). Super-resolution imaging was obtained using the tau-STED capability. For all live-cell imaging, cells were cultured in standard conditions.

Nikon TiE inverted platform coupled with TIRF E Motorized Illuminator, 100x/1.49 NA TIRF objective, and Dual-view2 camera splitter with green/red emission set was used to acquire TIRF imaging. Excitation of fluorophores 488 nm (GFP) and 561 nm (mCherry) were used. Image analysis was done using Fiji software. Huygens Software: by Scientific Volume, Imaging was used for deconvolution tasks and colocalization analysis. To analyze TIRFM images in Fiji, ROIs were drawn at the highest 488-nm signal regions (contact site) and the periphery of the cell. Integrated density values were measured throughout the movie lengths and ratios were calculated. All representative images were selected based on representative morphology, fluorescence levels, and optimal signal-to-noise ratio of the cohort.

## Statistical analysis

Statistical analysis and generation of graphs and plots were accomplished using Prism 8 (GraphPad). Two-way ANOVA with the Bonferroni test was applied for multiple comparisons. The level of statistical significance is represented by *$P < 0.0332$, **$P < 0.0021$, ***$P < 0.0002$, ****$P < 0.0001$. Data are shown as

mean ± standard error of the mean (SEM) unless otherwise specified.

## Data availability

This study includes no data deposited in external repositories.

## Peer review information

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

## Acknowledgements

We would like to thank Emilia Scharrig, Margaret McCormick, Waldo Spessott,
and Maria L. Sanmillan for their technical assistance and training. We thank
Thomas Jefferson University Office for Professional Writing, Publishing &
Communication for editing this manuscript. In addition, we thank Gerry
Hammond and Tamas Balla for the EGFP-C1ab-Prkd1 bio probe for DAG
detection. This work was supported by R01GM123020 grant from NIGMS to
CGG and T32 AI134646-4 grant from NIAID to NB. We would like to thank the
Bioimaging Shared Resource of the Sidney Kimmel Cancer Center and the Flow
Cytometry and Cell Sorting Facility (NCI 5 P30 CA-56036).

## Author contributions

**Nathalia Benavides**: Conceptualization; Data curation; Formal analysis;
Investigation; Visualization; Methodology; Writing—original draft; Writing—review
and editing. **Claudio G Giraudo**: Conceptualization; Data curation; Formal analysis;
Supervision; Funding acquisition; Investigation; Visualization; Methodology; Writing
—original draft; Project administration; Writing—review and editing.

## Disclosure and competing interests statement

The authors declare no competing interests.

# Expanded View Figures

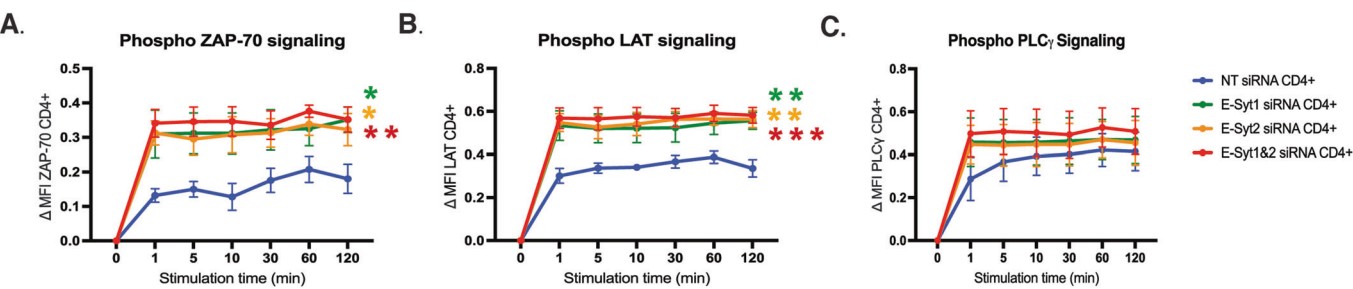

**Figure EV1.   E-Syt1 and -2 knockdown in primary CD4+ T cells enhances TCR signaling kinetics.**

(A–C) Flow cytometry analysis of CD3+CD4+ gated T cells from human PBMCs using signaling events measured by phosphorylation levels of early TCR signaling proteins ZAP70 at Y292 (A) LAT at Y226 (B) and PLC-γ1 at Y783 (C) upon stimulation with anti-CD3 and CD28 antibody-coated beads for indicated times. Plots show the change in median fluorescence intensity (MFI) between the indicated time of stimulation and $t = 0$. Plots are a mean ± SEM of at least three independent experiments. Significance was determined by two-way ANOVA with Bonferroni's multiple comparisons test against control NT siRNA CD4+ T cells (*$P < 0.0332$, **$P < 0.0021$, ***$P < 0.0002$). Source data are available online for this figure.

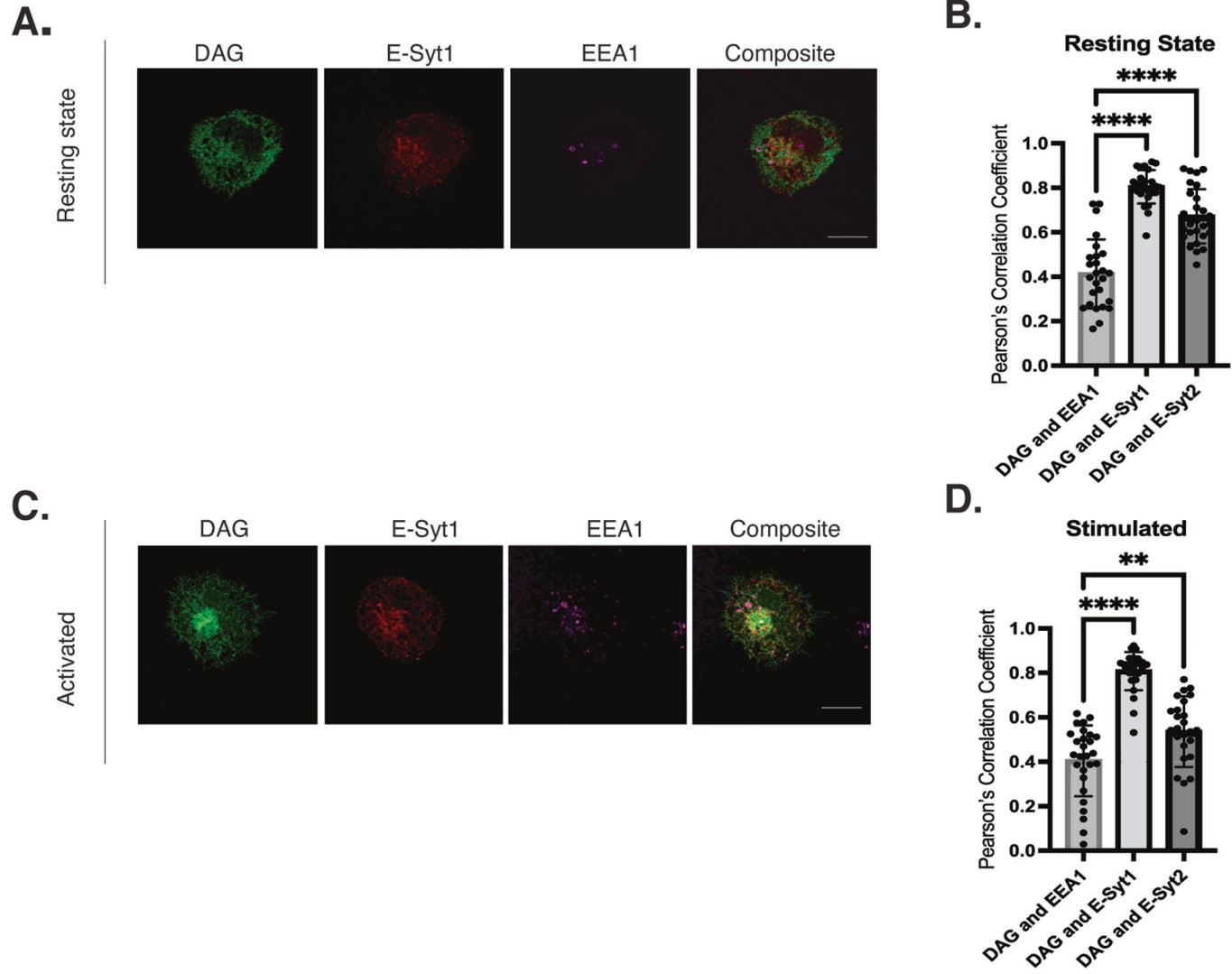

**Figure EV2. E-Syt1 and E-Syt2 colocalize with DAG in Jurkat cells.**

(A) Representative confocal images of WT-Jurkat cells transfected with GFP-PKCg-C1 probe (DAG) and stained with anti-E-Syt1 (red) and anti-EEA1 (Early Endosome Antigen 1, magenta) attached onto poly-lysine treated glass coverslips (resting state). Scale bars: 5 μm. (B) Pearson's correlation coefficient analysis of DAG-E-Syt1, DAG-E-Syt2, and DAG-EEA1 colocalization of all cell lines in resting state (DAG-EEA1 $n = 26$ cell, DAG-E-Syt1 $n = 26$ cells, DAG-E-Syt2 $n = 28$ cells) Plots are the mean ± SEM of at least three independent experiments. Significance was determined by one-way ANOVA with Bonferroni's multiple comparisons test against control WT-Jurkat cells (****$P < 0.0001$). (C) Representative confocal images of WT-Jurkat cells transfected with GFP-PKCg-C1 probe (DAG) and stained with anti-E-Syt1 (red) and anti-EEA1 (Early Endosome Antigen 1, magenta) attached onto poly-lysine treated glass coverslips plus anti-CD3 and -CD28 antibodies treated glass coverslips (activated state). Scale bars: 5 μm. (D) Pearson's correlation coefficient analysis of DAG-E-Syt1, DAG-E-Syt2, and DAG-EEA1 colocalization of all cell lines in stimulated conditions (DAG-EEA1 $n = 27$ cell, DAG-E-Syt1 $n = 27$ cells, DAG-E-Syt2 $n = 25$ cells). Plots are the mean ± SEM of at least three independent experiments. Significance was determined by one-way ANOVA with Bonferroni's multiple comparisons test against control WT-Jurkat cells (**$P < 0.0021$, ****$P < 0.0001$). Source data are available online for this figure.

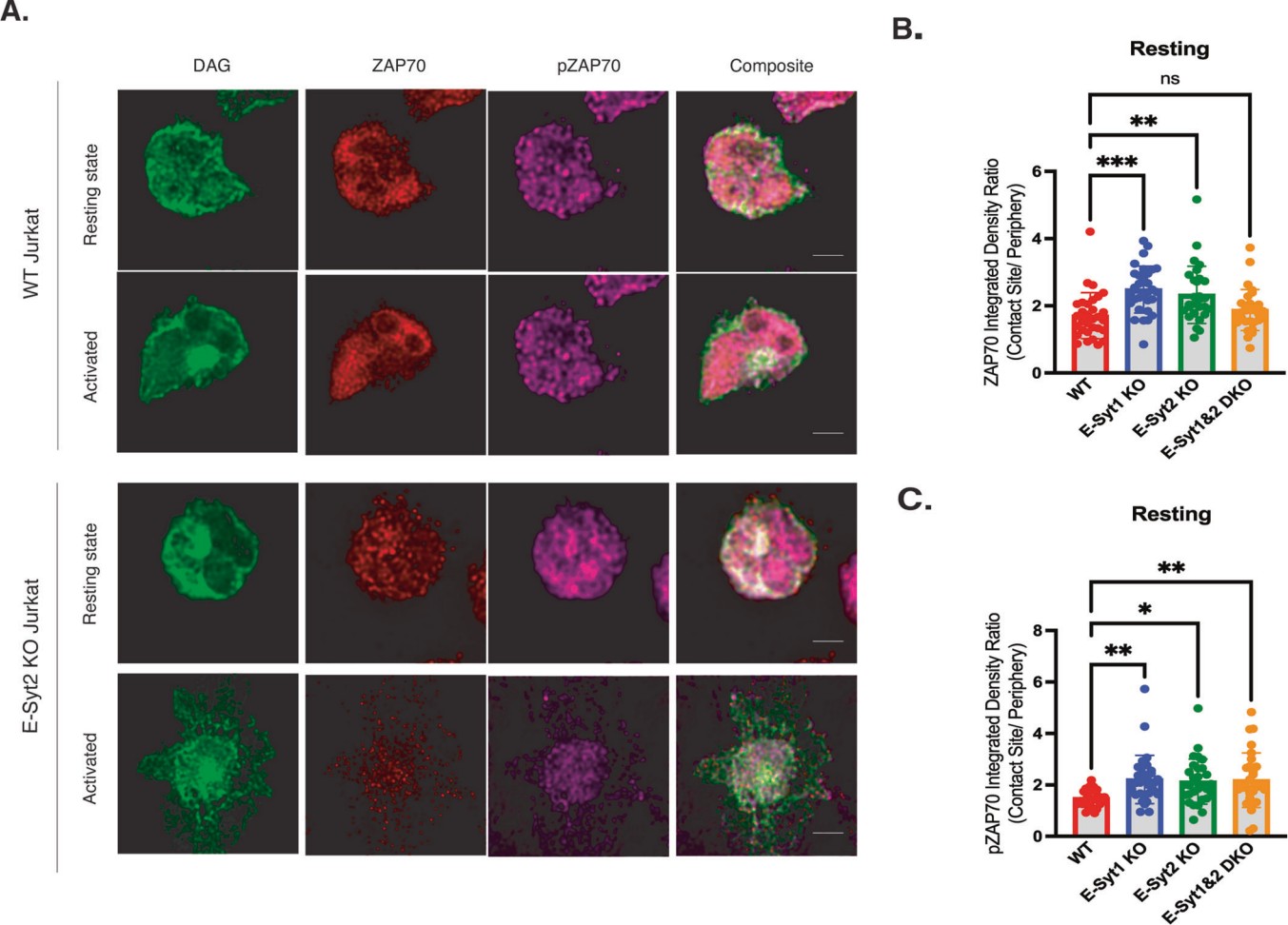

**Figure EV3. E-Syt proteins modulate p-ZAP70 and total-ZAP70 levels in Jurkat cells in resting conditions.**

(A) Representative WT and E-Syt2 KO Jurkat cells transfected with GFP-PKCg-C1 probe (DAG) and stained with anti-ZAP70 and anti-p-ZAP70 antibodies as signaling markers in resting and activated states. Scale bars: 5 µm. (B, C) Quantification of the ratio of the integrated fluorescent density of ZAP70 (B) p-ZAP70 (C) between the contact site area (DAG-rich area) and the cell periphery (low DAG signal). Data are shown as mean ± SEM of at least three independent experiments (WT $n = 31$ cells; E-Syt1 KO $n = 31$ cells; E-Syt2 KO $n = 29$ cells; E-Syt1&2 KO $n = 30$ cells). Significance was determined by one-way ANOVA with Bonferroni's multiple comparisons test against control WT Jurkat (*$P < 0.0332$, **$P < 0.0021$, ***$P < 0.0002$). Source data are available online for this figure.

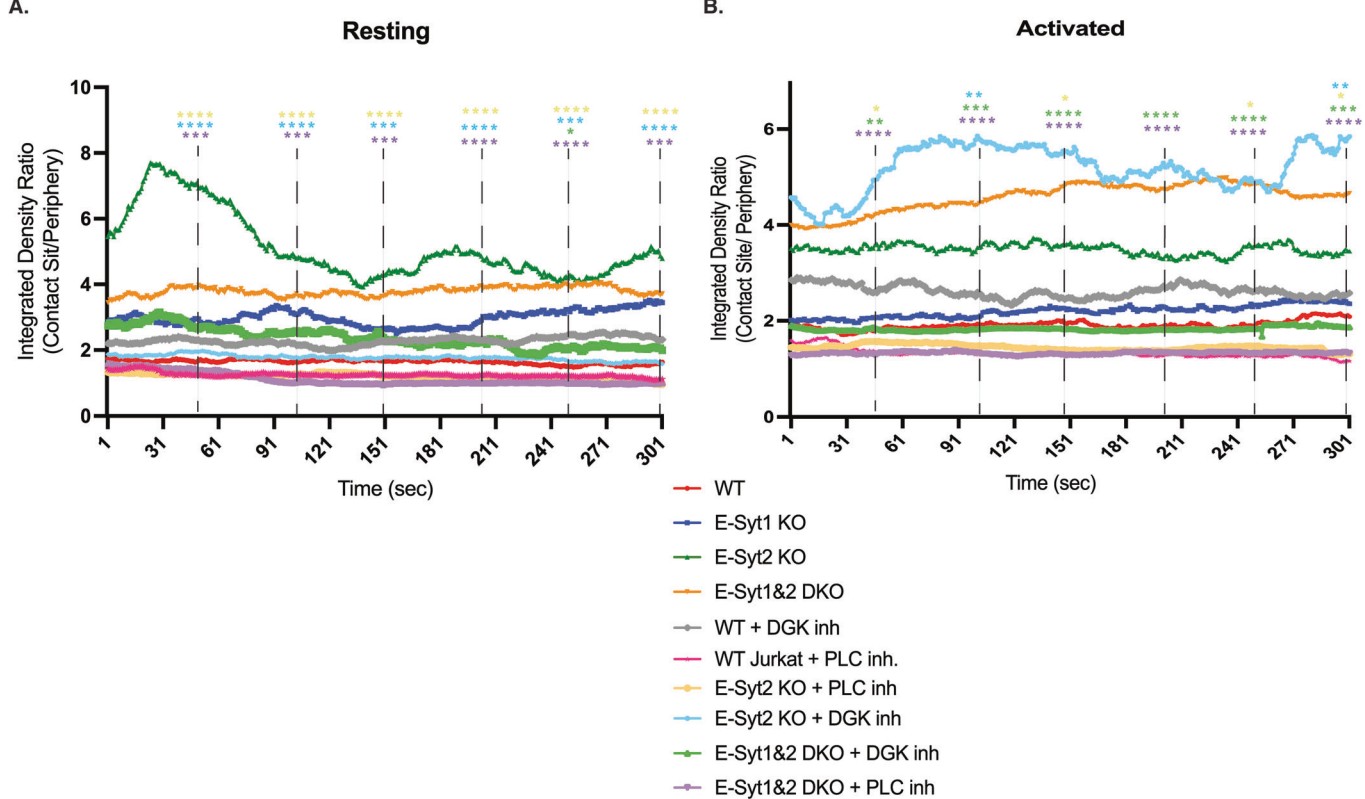

**Figure EV4. DAG production at the PM of Jurkat cells requires PLCγ activity.**

(A, B) Quantification from TIRF movies of WT and E-Syt KO cell lines transfected with GFP-C1ab for PM DAG detection treated over night with 5 µM PLCγ inhibitor or 300 µM DGK inhibitor in resting (A) or activated conditions (B). Quantification of the ratio of the integrated fluorescent density of DAG-rich areas and the periphery. Data are shown as mean ± SEM of at least three independent experiments (WT $n = 8$ cells; E-Syt1 KO $n = 8$ cells; E-Syt2 KO $n = 8$ cells, E-Syt1&2 KO $n = 8$ cells). Significance was determined by two-way ANOVA with Bonferroni's multiple comparisons test against the respective non-treated cell line control (*$P < 0.0332$, **$P < 0.0021$, ***$P < 0.0002$, ****$P < 0.0001$). For clarity, we only showed statistically significant points at the indicated times (50, 100, 150, 200, 250, 300 s). Source data are available online for this figure.

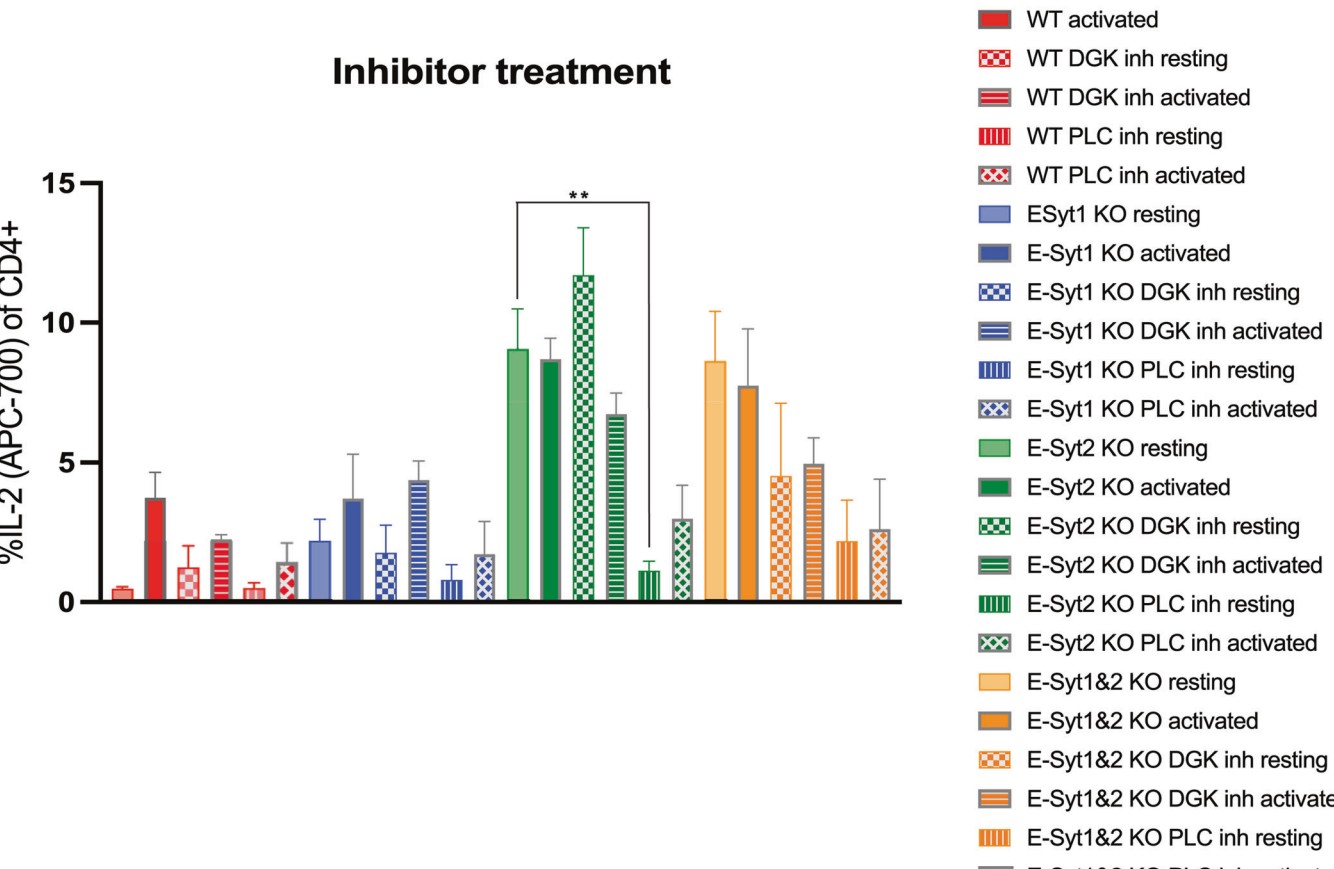

**Figure EV5.   Treatment of Jurkat cells with PLCγ inhibitor decreases IL-2 production regardless of E-Syt proteins.**

Intracellular staining of IL-2 measured by flow cytometry in WT and E-Syt KO Jurkat cell lines at resting state (faded colored bars). Jurkat cell lines stimulated with anti-CD3 and -CD28 coated beads for 12 h (solid colored bars). Jurkat cell lines were treated 5 h prior with 5 µM PLCγ inhibitor or 300 µM before starting the stimulation process. Figure legend indicates cell line, inhibitor treatment, and activation status. The data shown represent the mean ± SEM of at least four independent experiments. Significance was determined by one-way ANOVA with Bonferroni's multiple comparisons test against the respective non-treated cell line control (**$P < 0.0021$). Source data are available online for this figure.

