## [Peer Review File · EMBO Reports]

Extended-Synaptotagmin-1 and -2 Control T cell Signaling and Function

Nathalia Benavides and Claudio Giraudo

DOI: 10.15252/embr.202357350

Corresponding author(s): Claudio Giraudo (claudio.giraudo@jefferson.edu)

Review Timeline:

Submission Date:	17th Apr 23
Editorial Decision:	31st May 23
Revision Received:	30th Aug 23
Editorial Decision:	10th Oct 23
Revision Received:	22nd Oct 23
Accepted:	13th Nov 23

Transaction Report:

Dear Dr. Giraudo

Thank you for the submission of your research manuscript to our journal. We have now received the full set of referee reports that is copied below.

As you will see, the referees acknowledge that the findings are potentially interesting, but they also raise a number of concerns, which need to be addressed in a revision.

Referee 1 and 3 both pointed out that plating cells on coated slides does not really reproduce the IS. After a further discussion with referee 1, we think that no further experiments with a more physiological model are required but the limitation of the model needs to be discussed and the referee advises to "[...] avoid referring to the IS particularly when mentioning resting cells (plated onto poly-L lys). I believe that resting/activating conditions reflects better the reality."

Given these constructive comments, we would like to invite you to revise your manuscript with the understanding that the referee concerns (as detailed above and in their reports) must be fully addressed and their suggestions taken on board. Please address all referee concerns in a complete point-by-point response. Acceptance of the manuscript will depend on a positive outcome of a second round of review. It is EMBO reports policy to allow a single round of revision only and acceptance or rejection of the manuscript will therefore depend on the completeness of your responses included in the next, final version of the manuscript.

We realize that it is difficult to revise to a specific deadline. In the interest of protecting the conceptual advance provided by the work, we recommend a revision within 3 months (August 30th). Please discuss the revision progress ahead of this time with the editor if you require more time to complete the revisions.

I am also happy to discuss the revision further via e-mail or a video call, if you wish.

*******IMPORTANT NOTE:**

We perform an initial quality control of all revised manuscripts before re-review. Your manuscript will FAIL this control and the handling will be DELAYED if the following APPLIES:

- 1) A data availability section providing access to data deposited in public databases is missing. If you have not deposited any data, please add a sentence to the data availability section that explains that.
- 2) Your manuscript contains statistics and error bars based on $n=2$. Please use scatter blots in these cases. No statistics should be calculated if $n=2$.

When submitting your revised manuscript, please carefully review the instructions that follow below. Failure to include requested items will delay the evaluation of your revision.*****

- 1) a .docx formatted version of the manuscript text (including legends for main figures, EV figures and tables). Please make sure that the changes are highlighted to be clearly visible.
- 2) individual production quality figure files as .eps, .tif, .jpg (one file per figure). Please download our Figure Preparation Guidelines (figure preparation pdf) from our Author Guidelines pages <https://www.embopress.org/page/journal/14693178/authorguide> for more info on how to prepare your figures.
- 3) a .docx formatted letter INCLUDING the reviewers' reports and your detailed point-by-point responses to their comments. As part of the EMBO Press transparent editorial process, the point-by-point response is part of the Review Process File (RPF), which will be published alongside your paper.
- 4) a complete author checklist, which you can download from our author guidelines (<<https://www.embopress.org/page/journal/14693178/authorguide>>). Please insert information in the checklist that is also

reflected in the manuscript. The completed author checklist will also be part of the RPF.

5) Please note that all corresponding authors are required to supply an ORCID ID for their name upon submission of a revised manuscript (<<https://orcid.org/>>). Please find instructions on how to link your ORCID ID to your account in our manuscript tracking system in our Author guidelines (<<https://www.embopress.org/page/journal/14693178/authorguide#authorshipguidelines>>)

6) We replaced Supplementary Information with Expanded View (EV) Figures and Tables that are collapsible/expandable online. A maximum of 5 EV Figures can be typeset. EV Figures should be cited as "Figure EV1, Figure EV2" etc... in the text and their respective legends should be included in the main text after the legends of regular figures.

7) Please note that a Data Availability section at the end of Materials and Methods is now mandatory. In case you have no data that requires deposition in a public database, please state so instead of refereeing to the database. See also < <https://www.embopress.org/page/journal/14693178/authorguide#dataavailability>>. Please note that the Data Availability Section is restricted to new primary data that are part of this study.

Additional information on source data and instruction on how to label the files are available <<https://www.embopress.org/page/journal/14693178/authorguide#sourcedata>>.

10) Figure legends and data quantification:
The following points must be specified in each figure legend:

- the name of the statistical test used to generate error bars and P values,
- the number (n) of independent experiments (please specify technical or biological replicates) underlying each data point,
- the nature of the bars and error bars (s.d., s.e.m.)

- If the data are obtained from n {less than or equal to} 5, show the individual data points in addition to the SD or SEM.
- If the data are obtained from n {less than or equal to} 2, use scatter blots showing the individual data points.

See also the guidelines for figure legend preparation:
<https://www.embopress.org/page/journal/14693178/authorguide#figureformat>

11) Our journal encourages inclusion of *data citations in the reference list* to directly cite datasets that were re-used and obtained from public databases. Data citations in the article text are distinct from normal bibliographical citations and should directly link to the database records from which the data can be accessed. In the main text, data citations are formatted as follows: "Data ref: Smith et al, 2001" or "Data ref: NCBI Sequence Read Archive PRJNA342805, 2017". In the Reference list, data citations must be labeled with "[DATASET]". A data reference must provide the database name, accession number/identifiers and a resolvable link to the landing page from which the data can be accessed at the end of the reference. Further instructions are available at <<https://www.embopress.org/page/journal/14693178/authorguide#referencesformat>>.

12) As part of the EMBO publication's Transparent Editorial Process, EMBO reports publishes online a Review Process File to accompany accepted manuscripts. This File will be published in conjunction with your paper and will include the referee reports,

your point-by-point response and all pertinent correspondence relating to the manuscript.

Yours sincerely,

Referee #1:

This is an interesting and innovative study where the authors address the relevance of the extended-synaptotagmin (E-Syst) protein family in the control of functional T cell responses. The E-Syst proteins are endoplasmic (ER) resident proteins that transfer glycerolipids between cell membranes in a PtdIns(4,5)P₂- and Ca²⁺-regulated mechanism. The main function ascribed to these proteins is to clear the plasma membrane of the diacylglycerol (DAG) generated upon receptor triggered PLC activation. Polarized DAG generation at the site of cell-cell contact (named immune synapse, IS) is of critical relevance for T cell functions that range from initiation of signals to the control of mTOC polarization and polarized secretion. The main enzymes reported to regulate DAG at the IS are the DGK family of proteins. No studies to this date have investigated the contribution of E-Syst to DAG homeostasis during T cell activation, so the purpose of this work is of interest by proposing additional mechanism in the control of DAG metabolism during antigen recognition. The authors demonstrate expression of the three members of the family and use silencing approaches to demonstrate that silencing of individual E-Syst isoforms 1 or 2 or both affects distinct aspects of T cell functions. To this end they first use primary human CD8 T cells to show a slight effect in the cytotoxic functions and the Jurkat T cell model to demonstrate enhanced ERK activation. All these functions could be attributed to enhanced DAG functions so then they use imaging techniques to investigate the dynamics of DAG generation in resting and activated Jurkat cells. The experiments are well executed and it is clear that genetic silencing of E-Syst proteins enhances initial signals and alters DAG distribution. However, there are some important issues that need clarification and some conclusions that cannot be deduced from the experiments as they are presented. Some of the main questions are

1. The more relevant issue refers to the definition of IS and the main claim stated in the title and along the manuscript. The IS represents the zone where the T cell receptor and co-stimulatory molecules engage with antigens that are presented by professional antigen presenting cells (APC). As the authors indicate DAG is accumulated at this cell-cell contact zone first by PLC activation and then by continuous trafficking between DAG-enriched Golgi and recycling endosomes and the plasma membrane. In this study the authors activate the cells by plating them onto antibody-coated slides. Albeit extensively used as a model of T cell activation this approach does not reproduce the formation of a bona fide IS that requires contact between two cells and membrane reorganization based on TCR and integrin interactions. When plated onto antibody-coated slides the cells acquire a spread "fried egg" form with active traffic from and to the PM and internal localizations and close contact of Golgi and ER to the slide contact site that facilitates the accumulation observed for DAG at the center of the cell. But the continuous reference to the IS is not correct as this model does not reproduce the classical bull eyed distribution sustained by integrins and components of the TCR. Even more, referring to IS when cells are plated onto poly-L-Lysine (resting conditions) is not accurate, as resting conditions correspond to a situation where T cells are not in contact with APCs so there is no IS.
2. Functional experiments in Fig 1, 2 and 3 suggest that E-Syst 2 is the predominant isoform limiting T cell functions. On the other hand, experiments in Fig 4 show similar abnormal distribution of DAG in cells silenced for one or the other isoform. In all cases individual or simultaneous silencing shows enhanced DAG accumulation in some area (Golgi?) in resting conditions that could then explain enhanced DAG accumulation upon activation. Instead of using LAT it would help to use a phosphoLAT antibody or some other marker that could provide a hint about why silencing of E-Syst 2 is the one that provides enhanced signals.
3. The PM localization of E-Syst1 and E-Syst2 in Jurkat at resting state is quite different from that reported for other cells where E-Systs are ER proteins that contact the PM in response to PIP₂ hydrolysis by PLC and Calcium generation. The PM is difficult to

discriminate from the ER in T cells that contain a very large nuclei with small cytosolic space. Additional markers for nuclei, ER and PM would help to discriminate. In the top panel of figure 4A the difference between resting (small and round) and activated (flat and spread) Jurkat cells is clearly observed with marked recruitment of E-St1 to internal localizations that nicely correlate with DAG and CD4. However, the cell selected in the lower panel does not seem to be fully activated or the analyzed plane is different. It is difficult to see differences between one isoform and the other since DAG and CD4 distribution is so different.

4. Experiments in Fig4B and Fig 5 strongly suggest that silencing of E-Syst proteins results in altered DAG accumulation under resting conditions. The authors acknowledge this by stating "in the resting state, all E-Syt KO cell lines displayed a large patch of DAG accumulation at the IS membrane". As already indicated in resting state there is no IS so DAG is accumulating either at the plasma membrane even under conditions of no receptor activation. Plating the cells onto poly-L-Lysine enhanced calcium so this could be providing some basal signal able to sustain E-Syst under resting stimulation. Basal DAG accumulation could facilitate some basal signal that could enhance those triggered by receptor stimulation. This has been shown for instance for DGKzeta silencing.

Overall, the authors need to clarify the consequences of silencing E-Syts in the Jurkat system and discuss the differences with what is known in other systems. The main role of these proteins is to clear the PM of diacylglycerol (DAG) formed after receptor-triggered activation of PLC. Genome-edited cells lacking E-Syts in general do not exhibit abnormalities at rest, but show enhanced and sustained accumulation of PM diacylglycerol following PtdIns(4,5)P₂ hydrolysis by PLC activation. In this system the authors observe DAG accumulation in resting cells suggesting that E-Syts are important in basal conditions but this issue must be examined in more detail.

Additional comments

5. Enhanced pERK fits well with defects in DAG metabolism but the enhanced activation of Tyrosine phosphorylated LAT and ZAP70 as it is shown in Fig 2 it is difficult to reconcile with enhanced DAG functions. Any explanation for this? ZAP70 and LAT are two targets of Lck that is bound to CD4, could this be the result of enhanced CD4 clustering? Analysis of pCD3 and/or Lck would help to clarify this.

6. In figure 3 the authors determine IL2 production by FACS analysis to demonstrate the consequences of silencing the distinct E-Syst isoforms. For activation the time selected is that of 3 hours after stimulation that seems too short to determine transcriptional activation and IL2 production. In fact in WT cells only 5% of the cells are IL2 positive with almost no changes in the MFI (as shown in EV3). E-Syst silencing results in no changes in IL2 MFI (suggesting that on a per cell basis there is almost no IL2 production) and the increase on % corresponds to that observed in resting conditions. Experiments should be repeated at longer times (24 hours) where cells are fully activated to conclude that silencing of E-Syst has a real contribution on T cell activation.

Referee #2:

The manuscript "Extended-Synaptotagmin-1 and -2 Control Signaling and Functionality of T Cells at the Immune Synapse", is the first description of a novel role of E-Syts in DAG dynamics at the immune synapse. The concept that mechanisms of lipid regulation at the synaptic membrane via lipid transport play roles in TCR signaling is an interesting notion. The data support the author's central finding. One important point that is not very well addressed is how E-Syts regulate DAG dynamics in T cells. It is not clear if the increase of DAG occurs at the expense of PI(4,5)P₂ cleavage or is mediated by transport of DAG. Are diacylglycerol kinases or other molecules involved in this process? In addition, it could be very interesting to see ER-PM contact sites at the IS.

The authors used primary CD8⁺ and Jurkat T cells, key experiments should be performed also in primary CD4⁺ cells.

Please see other more specific points that should be addressed:

Lines 46-51: the authors introduce the immune synapse, but they describe lytic synapse, please clarify.

Line 106: please justify the interaction of E-Syt1 with the vesicular trafficking regulator Syntaxin 11.

Line 115: "Knockdown efficiency was tested by WB", please indicate the quantification of KD efficiency.

Lines 111-130: it could be interesting to analyse the polarization of lytic granules and the localization of DAG at the lytic synapse. Is statistically significant the increase in degranulation in unstimulated KD CD8⁺ T cells showed in figure 1B?

Lines 145-160: the data are presented as change in median fluorescence intensity between the indicated time of stimulation and t=0, there are differences in basal levels of phosphorylation of ZAP-70, LAT and PLCγ1 as suggested by IL2 production?

Lines 213-214: there is any effect of E-Syt protein overexpression? Please show DAG levels at the immune synapse in control

cells.

Lines 220-238: Levels of PM DAG should be very low in resting cells and transiently increase upon activation, however in the images it seems that DAG is present also in unstimulated cells. Please explain. It could be very interesting to look at z-stack of T cell to see the accumulation of DAG at the IS.

Lines 249-251: "E-Syt1&2 DKO cells displayed a stronger DAG signal than the single E-Syt KO cells, but not in a well-organized manner as previously observed in activated WT cells". Please quantify.

Lines 296-298: "Unlike our previous results in resting state conditions which showed that only E-Syt2 KO cells displayed a spike in DAG accumulation within the first few minutes of contact with the surface", this is not clear from the showed graph. Why DAG levels are higher in resting than in activated E-Syt2KD cells?

Line 367: "We observed that in the absence of E-Syts, cells bypass TCR-activation and are in a pre-activated state, independent of standard stimulation". I am wondering if there is an induction of apoptosis after T cell stimulation (AICD, activation induced cell death). Is cell viability modulated by the expression of E-Syt proteins?

Line 703: Figure 4. Please increase the number of analysed cells and indicate the number of performed experiments.

Line 801: Expanded View Figure 2. The WB anti-LAT seems to be different in control and KD cells, please verify.

Line 869: Expanded View Figure 6. Please refer to this figure in the text. Control and E-Syt KO cells should be represented in unstimulated and stimulated cells.

Referee #3:

The manuscript titled "Extended-Synaptotagmin-1 and -2 Control Signaling and Functionality of T Cells at the Immune Synapse" adds to growing body of evidence for the roles of ESYT proteins in T cell activation. In particular, the manuscript claims to show an inhibitory role of ESYT1 and ESYT2 in (Jurkat) T cell TCR signaling through diacylglycerol dynamics at the interacting cell membrane. Strengths of the work include a novel mechanistic concept and robust experimental and analytical methods including knock-down, knock-out, fluorescent protein and immunofluorescence microscopy including TIRF. Weaknesses include predominant use of Jurkat cell as the cellular model of TCR synaptic function and limited modeling of the immunological synapse using CD3/CD28-coated support surface. Nonetheless, the important overall conclusion that ESYT1/2 represent potential targets for increasing T cell function is acceptable based on the results shown in Figure 1.

1. Does this manuscript report a single key finding? YES

Demonstrating an inhibitory effect of ESYT1/2 on T cell degranulation.

2. Is the reported work of significance (YES), or does it describe a confirmatory finding or one that has already been documented using other methods or in other organisms etc (NO)? YES, the report contains a substantial element of novelty in terms of the mechanistic insights into DAG.

3. Is it of general interest to the molecular biology community? YES, the report is of interest for the fields of T cell immunology and T cell-based immunotherapy.

4. Is the single major finding robustly documented using independent lines of experimental evidence (YES), or is it really just a preliminary report requiring significant further data to become convincing, and thus more suited to a longer –format article (NO)? YES, the findings are robust with regard to the use of advanced analytical methods including cytotoxicity increase upon knockdown in primary CD8 T cells. However, the relevance of mechanistic claims for natural T cells is uncertain due to overreliance on the Jurkat cell system for the mechanistic studies. Extending the mechanistic findings onto primary T cells with the use of MHC ligands for TCR activation would address this concern.

Additional points:

128: It is difficult to agree with the notion of E-Syt1 and E-Syt2 independency based on the finding that E-Syt1&2 double-knockdown did not produce an additive effect. A more fitting interpretation of this particular result would be that the roles are interdependent in a consecutive manner. If the roles were truly independent such as in disconnected parallel pathways, double knockdown would be expected to produce an additive effect.

267: Please clarify the sentence: "Given our previous findings that lack of expression of E-Syt2 alone in conjunction with E-Syt1 induces DAG accumulation at the PM, ..." Is it alone or in conjunction?

801: Phospho instead of phosphor.

Figure legend titles for figures 2, 3, 4, 5 and extended figures 2 and 3 are misleading by stating T cells. The titles should clearly state Jurkat cells.

Figure 1 legend title should refer to both E-Syt1 and E-Syt2 (currently only E-Syt1).

Fig 1C is difficult to decipher - please consider enlarging this graph.

Fig 4 Color scheme is confusing by showing CD4 using the purple color for WT cells and the red color for KO cells. It would be better to use the same color. This experiment should not be divided into panels A and B given the groups are analyzed against each other (C, D).

Response to the Reviewers

Referee #1:

This is an interesting and innovative study where the authors address the relevance of the extended-synaptotagmin (E-Syst) protein family in the control of functional T cell responses. The E-Syst proteins are endoplasmic (ER) resident proteins that transfer glycerolipids between cell membranes in a PtdIns(4,5)P₂- and Ca²⁺-regulated mechanism. The main function ascribed to these proteins is to clear the plasma membrane of the diacylglycerol (DAG) generated upon receptor triggered PLC activation. Polarized DAG generation at the site of cell-cell contact (named immune synapse, IS) is of critical relevance for T cell functions that range from initiation of signals to the control of mTOC polarization and polarized secretion. The main enzymes reported to regulate DAG at the IS are the DGK family of proteins. No studies to this date have investigated the contribution of E-Syst to DAG homeostasis during T cell activation, so the purpose of this work is of interest by proposing additional mechanism in the control of DAG metabolism during antigen recognition. The authors demonstrate expression of the three members of the family and use silencing approaches to demonstrate that silencing of individual E-Syst isoforms 1 or 2 or both affects distinct aspects of T cell functions. To this end they first use primary human CD8 T cells to show a slight effect in the cytotoxic functions and the Jurkat T cell model to demonstrate enhanced ERK activation. All these functions could be attributed to enhanced DAG functions so then they use imaging techniques to investigate the dynamics of DAG generation in resting and activated Jurkat cells.

The experiments are well executed and it is clear that genetic silencing of E-Syst proteins enhances initial signals and alters DAG distribution. However, there are some important issues that need clarification and some conclusions that cannot be deduced from the experiments as they are presented.

Some of the main questions are

1. The more relevant issue refers to the definition of IS and the main claim stated in the title and along the manuscript. The IS represents the zone where the T cell receptor and co-stimulatory molecules engage with antigens that are presented by professional antigen presenting cells (APC). As the authors indicate DAG is accumulated at this cell-cell contact zone first by PLC activation and then by continuous trafficking between DAG-enriched Golgi and recycling endosomes and the plasma membrane. In this study the authors activate the cells by plating them onto antibody-coated slides. Albeit extensively used as a model of T cell activation this approach does not reproduce the formation of a bona fide IS that requires contact between two cells and membrane reorganization based on TCR and integrin interactions. When plated onto antibody-coated slides the cells acquire a spread

"fried egg" form with active traffic from and to the PM and internal localizations and close contact of Golgi and ER to the slide contact site that facilitates the accumulation observed for DAG at the center of the cell. But the continuous reference to the IS is not correct as this model does not reproduce the classical bull eyed distribution sustained by integrins and components of the TCR. Even more, referring to IS when cells are plated onto poly-L-Lysine (resting conditions) is not accurate, as resting conditions correspond to a situation where T cells are not in contact with APCs so there is no IS.

Answer: We agree with the reviewer's comment that the terminology "IS" is not appropriate in our models, especially when cells are plated onto poly-L-lysine coated plates in resting conditions. For this reason we have modified our language in the text and clarified that the DAG we are visualizing is at the plasma membrane.

2. Functional experiments in Fig 1, 2 and 3 suggest that E-Syst 2 is the predominant isoform limiting T cell functions. On the other hand, experiments in Fig 4 show similar abnormal distribution of DAG in cells silenced for one or the other isoform. In all cases individual or simultaneous silencing shows enhanced DAG accumulation in some area (Golgi?) in resting conditions that could then explain enhanced DAG accumulation upon activation. Instead of using LAT it would help to use a phosphoLAT antibody or some other marker that could provide a hint about why silencing of E-Syst 2 is the one that provides enhanced signals.

Answer: We appreciate the reviewer's suggestion to use another marker that could help us elucidate why the downregulation of E-Syt2 enhances T-cell signaling. For this, we imaged our Jurkat cell lines with phospho-LAT as well as phospho-ZAP and ZAP together and quantified the integrated density signal of each marker at the PM in both resting and activated conditions. Our new data in Figure 4B-E and Fig. EV8, show that indeed, the absence of E-Syt1&2 increases the amount of quantified signal of these key signaling proteins.

3. The PM localization of E-Syst1 and E-Syst2 in Jurkat at resting state is quite different from that reported for other cells where E-Syts are ER proteins that contact the PM in response to PIP2 hydrolysis by PLC and Calcium generation. The PM is difficult to discriminate from the ER in T cells that contain a very large nuclei with small cytosolic space. Additional markers for nuclei, ER and PM would help to discriminate. In the top panel of figure 4A the difference between resting (small and round) and activated (flat and spread) Jurkat cells is clearly observed with marked recruitment of E-St1 to internal localizations that nicely correlate with DAG and CD4. However, the cell selected in the lower panel does not seem to be fully activated or the analyzed plane is different. It is difficult to see differences between one isoform and the other since DAG and CD4 distribution is so different.

Answer: We agree with the reviewer that it is difficult to discriminate ER localization from PM, especially in rounded cells like T-cells. For this reason, we have performed additional imaging studies comparing the localization of either ESyt1 or ESyt2 with different subcellular markers, KDEL-receptor as ER membrane, Cell Mask dye as PM, and DRAQ5 as nuclear staining in resting and active conditions (Fig. EV6). These data

show that distribution of both ESyt1 and 2 mainly resemble cell mask stain and to a lesser extent ER staining.

4. Experiments in Fig4B and Fig 5 strongly suggest that silencing of E-Syt proteins results in altered DAG accumulation under resting conditions. The authors acknowledge this by stating "in the resting state, all E-Syt KO cell lines displayed a large patch of DAG accumulation at the IS membrane". As already indicated in resting state there is no IS so DAG is accumulating either at the plasma membrane even under conditions of no receptor activation. Plating the cells onto poly-L-Lysine enhanced calcium so this could be providing some basal signal able to sustain E-Syt under resting stimulation. Basal DAG accumulation could facilitate some basal signal that could enhance those triggered by receptor stimulation. This has been shown for instance for DGKzeta silencing.

Answer: We concur with the reviewers remark that even cells plated onto poly-L-Lysine coverslips produce a basal level of DAG at the PM as it is now shown in the new 3D reconstruction images provided in Fig. 4 F, G. As we stated before, we have modified the text throughout the manuscript replacing "IS membrane" with "PM".

Overall, the authors need to clarify the consequences of silencing E-Syts in the Jurkat system and discuss the differences with what is known in other systems. The main role of these proteins is to clear the PM of diacylglycerol (DAG) formed after receptor-triggered activation of PLC. Genome-edited cells lacking E-Syts in general do not exhibit abnormalities at rest, but show enhanced and sustained accumulation of PM diacylglycerol following PtdIns(4,5)P₂ hydrolysis by PLC activation. In this system the authors observe DAG accumulation in resting cells suggesting that E-Syts are important in basal conditions but his issue must be examined in more detail.

Answer: We agree with reviewer 2 , that the accumulation of DAG in resting conditions is an interesting phenotype. We further examined the functional consequences of increased DAG in different cellular processes. We did not find any significant increase in cell death induced by the absence of ESyt proteins (Fig EV. 13); however, we found increased IL-2 production as shown in Fig. 3. Moreover, experiments using PLC γ and DGK inhibitors showed that the increased accumulation of DAG in resting condition is also dependent on PLC γ activity (Fig. EV 11 and 12).

Additional comments

5. Enhanced pERK fits well with defects in DAG metabolism but the enhanced activation of Tyrosine phosphorylated LAT and ZAP70 as it is shown in Fig 2 it is difficult to reconcile with enhanced DAG functions. Any explanation for this?. ZAP70 and LAT are two targets of Lck that is bound to CD4, could this be the result of enhanced CD4 clustering?. Analysis of pCD3 and/or Lck would help to clarify this.

Answer: We are also intrigued by how DAG accumulation in E-Syt KO/KD cells induces phosphorylation of upstream signaling molecules like LAT and ZAP70. New Imaging studies showing accumulation of pLAT and pZAP70 with CD4 in resting condition (New Fig. 4B-E), thus providing some evidence for a DAG-induced clustering effect nucleating CD4 and the signaling molecules as the reviewers suggested. We have now discussed this possibility in the text .

6. In figure 3 the authors determine IL2 production by FACS analysis to demonstrate the consequences of silencing the distinct E-Syst isoforms. For activation the time selected is that of 3 hours after stimulation that seems too short to determine transcriptional activation and IL2 production. In fact in WT cells only 5% of the cells are IL2 positive with almost no changes in the MFI (as shown in EV3). E-Syst silencing results in no changes in IL2 MFI (suggesting that on a per cell basis there is almost no IL2 production) and the increase on % corresponds to that observed in resting conditions. Experiments should be repeated at longer times (24 hours) where cells are fully activated to conclude that silencing of E-Syst has a real contribution on T cell activation.

Answer: We want to apologize for the confusion in the section describing the cytokine production assay. To clarify this issue, we want to point out that the cells were activated with anti-CD3/28 coated beads for 3 h, followed by the addition of the Golgi stop reagent and further incubated for another 12 h more. Cells were then stained and analyzed by Flow Cytometry. We agree with the reviewer that the IL2 production of Jurkat WT cells is relatively low. We have tried longer activation times 24-30h and different activation surfaces but in our hands Jurkat cells behave as poor IL-2 producers upon engaging CD3-CD28 receptors. However they produce significant amount IL2 upon PMA and Ionomycin stimulation (positive control (Fig 3C). Nonetheless, we have performed similar IL2 production experiments with primary CD4+ and CD8+ ESyt-KD cells as the reviewer 3 suggested and we have observed that ESyts-KD produce significantly more IL2 than NT siRNA control cells in resting conditions. (New Fig. 3 D,E)

Referee #2:

The manuscript "Extended-Synaptotagmin-1 and -2 Control Signaling and Functionality of T Cells at the Immune Synapse", is the first description of a novel role of E-Syts in DAG dynamics at the immune synapse. The concept that mechanisms of lipid regulation at the

synaptic membrane via lipid transport play roles in TCR signaling is an interesting notion. The data support the author's central finding. One important point that is not very well addressed is how E-Syts regulate DAG dynamics in T cells. It is not clear if the increase of DAG occurs at the expense of PI(4,5)P2 cleavage or is mediated by transport of DAG. Are diacylglycerol kinases or other molecules involved in this process? In addition, it could be very interesting to see ER-PM contact sites at the IS.

Answer: We appreciate these critical insights. To address this question we now included data from new experiments using either PLCg or DGK inhibitors showing that the accumulation of DAG at the PM of ESyt-KO cells depends upon PLCg activity (Fig EV 11, and 12). Interestingly, the addition of DGK inhibitor did not alter DAG levels in resting conditions but increased DAG at the PM in ESyt2-KO-activated cells. These data support a model in which ESyts control DAG levels at resting state while both ESyts and DGKs modulates DAG levels in activated conditions.

The authors used primary CD8+ and Jurkat T cells, key experiments should be performed also in primary CD4+ cells.

Answer: We appreciate the reviewer's suggestion to extend our studies to primary CD4+ T cells. We agree that key experiments should also be performed in primary T cells. To address this, we have conducted phospho-protein signaling kinetics assays in CD4+ and CD8+ T cells transfected with siRNA for E-Syt proteins (Figure 1D-F and Fig EV2.). Our new data suggests that E-Syt1&2 double knockdown enhances phospho-ZAP70 and LAT signaling kinetics in CD4 and CD8 primary T cells. Additionally, we measured the %IL-2 expressing CD4 and CD8 T cells and noticed an increase upon E-Syt2 knockdown and E-Syt1&2 double-knockdown respectively in the resting state (New Fig 3D, E).

Please see other more specific points that should be addressed:

Lines 46-51: the authors introduce the immune synapse, but they describe lytic synapse, please clarify.

Answer: We have modified this section of the introduction and described more broadly the Immune synapses not just lytic synapses.

Line 106: please justify the interaction of E-Syt1 with the vesicular trafficking regulator Syntaxin 11.

Answer: Syntaxin 11 is membrane fusion protein, and it is believed that mediated the release of lytic granule content at the IS. Interestingly, Syntaxin 11 is attached to the cell membrane through a lipid anchored-moiety. In our search for Syntaxin 11 interacting proteins using IP and MassSpec, ESyts proteins showed up as potential interactors. We hypothesize that Syntaxin 11 might localize in DAG-rich areas of the PM, which is in close proximity to ESyts proteins and where exocytosis usually occurs.

This special lipid microenvironment generated by DAG may favor the association between Syntaxin 11 and ESyt proteins. We have now included this information in the text.

Line 115: "Knockdown efficiency was tested by WB", please indicate the quantification of KD efficiency.

Answer: we apologize for not adding this information before. We have added the quantifications of KD efficiency in the new Fig 1a.

Lines 111-130: it could be interesting to analyse the polarization of lytic granules and the localization of DAG at the lytic synapse. Is statistically significant the increase in degranulation in unstimulated KD CD8+ T cells showed in figure 1B?

Answer: We appreciate this interesting comment. To address this, we have performed new imaging studies using ESyt KD human PBMCs and analyzed the colocalization of the lytic granule marker CD63 with DAG at the PM in either resting or activated conditions. These results revealed an increased colocalization of CD63 in ESYT KD cells compared with WT cells. This data is now included in new Fig. EV10.

Lines 145-160: the data are presented as change in median fluorescence intensity between the indicated time of stimulation and t=0, there are differences in basal levels of phosphorylation of ZAP-70, LAT and PLC γ 1 as suggested by IL2 production?

Answer: We understand the reviewer's concerns. To address this, we performed statistical analysis of the MFI of each marker at t=0 through different experiments and we have not found a statistically significant difference between cell lines. This data is provided as new Fig. EV3 E-G.

Lines 213-214: there is any effect of E-Syt protein overexpression? Please show DAG levels at the immune synapse in control cells.

Answer: To address this question we have quantified the total amount of DAG at the PM of cells transfected with either ESyt-1 or ESyt-2 using a microscopy approach. This data shows that there are no significant differences in the amount of DAG present at the PM of ESyt overexpressing cells compared with WT cells. This data is included in Fig. EV 9).

Lines 220-238: Levels of PM DAG should be very low in resting cells and transiently increase upon activation, however in the images it seems that DAG is present also in unstimulated cells. Please explain. It could be very interesting to look at z-stack of T cell to see the accumulation of DAG at the IS.

Answer: As reviewer 1 pointed out, even cells plated only onto poly-L-Lysine coated surfaces (non-stimulated conditions) enhanced calcium so this could be providing some basal signal able to induce the generation DAG, which in the absence of ESyt proteins leads to a noticeable DAG accumulation at the PM. We also provide new 3D

reconstruction images Fig. 4 F, G showing DAG presence at the PM in WT-cells resting condition, and increased DAG accumulation in ESyt2 KO cells resting conditions.

Lines 249-251: "E-Syt1&2 DKO cells displayed a stronger DAG signal than the single E-Syt KO cells, but not in a well-organized manner as previously observed in activated WT cells". Please quantify.

Answer: We apologize for not including this data before. To address this, we have quantified the integrated density ratio of DAG signal in all of our Jurkat cell lines and conditions. Our new data now in Figure 4C, shows that DAG signal is significantly higher in our E-Syt2 and E-Syt1&2 DKO cell lines. Thus further supporting our conclusion that E-Syt proteins modulate DAG levels in T cells.

Lines 296-298: "Unlike our previous results in resting state conditions which showed that only E-Syt2 KO cells displayed a spike in DAG accumulation within the first few minutes of contact with the surface", this is not clear from the showed graph. Why DAG levels are higher in resting than in activated E-Syt2KD cells?

Answer: To clarify this issue, we want to point out that the quantification in the graph shows the DAG fluorescent intensity ratio between the center of the contact point area and the cell periphery. In resting ESyt KO cells, the amount of DAG in the cell periphery is relatively low compared with the center, whereas in the activated conditions the total DAG levels significantly increase in both the center and periphery, thus resulting in a lower ratio. As mentioned above we have added the quantification of the total DAG measured as total integrated density (Fig. 5), showing a higher level of DAG in activated vs resting.

Line 367: "We observed that in the absence of E-Syts, cells bypass TCR-activation and are in a pre-activated state, independent of standard stimulation". I am wondering if there is an induction of apoptosis after T cell stimulation (AICD, activation induced cell death). Is cell viability modulated by the expression of E-Syt proteins?

Answer: We thank the reviewer for bringing up this important point. To address this question we performed an experiment to measure activation induced cell-death in WT and ESyt KO cell lines using a flow cytometry assay. Results from these experiments showed that ESyt KO cell lines did not show any statistically significant difference in cell death upon activation when compared with WT-cells. This data is now included in new Fig EV. 13.

Line 703: Figure 4. Please increase the number of analysed cells and indicate the number of performed experiments.

Answer: As suggested by the reviewer, we have performed additional experiments in Fig. EV7 B, C and increased the number of analyzed cells for these conditions (colocalization of DAG vs. CD4). For the resting conditions we have analyzed the following cells WT: 38, ESyt1KO: 39, ESyt2KO: 38, ESyt1&2 DKO:40. For the activated

conditions we have analyzed the following cells WT: 44, ESYT1KO: 41, ESYT2KO: 38, ESYT1&2 DKO:43.

Line 801: Expanded View Figure 2. The WB anti-LAT seems to be different in control and KD cells, please verify.

Answer: We apologize for this oversight. We have selected better representative anti-LAT WB images for the NT siRNA and E-Syt1&2 siRNA samples in Fig EV3.

Line 869: Expanded View Figure 6. Please refer to this figure in the text. Control and E-Syt KO cells should be represented in unstimulated and stimulated cells.

Answer: We have modified this figure by adding unstimulated and stimulated conditions we now cited the figure in the text accordingly.

Referee #3:

The manuscript titled "Extended-Synaptotagmin-1 and -2 Control Signaling and Functionality of T Cells at the Immune Synapse" adds to growing body of evidence for the roles of ESYT proteins in T cell activation. In particular, the manuscript claims to show an inhibitory role of ESYT1 and ESYT2 in (Jurkat) T cell TCR signaling through diacylglycerol dynamics at the interacting cell membrane. Strengths of the work include a novel mechanistic concept and robust experimental and analytical methods including knock-down, knock-out, fluorescent protein and immunofluorescence microscopy including TIRF. Weaknesses include predominant use of Jurkat cell as the cellular model of TCR synaptic function and limited modeling of the immunological synapse using CD3/CD28-coated support surface. Nonetheless, the important overall conclusion that ESYT1/2 represent potential targets for increasing T cell function is acceptable based on the results shown in Figure 1.

1. Does this manuscript report a single key finding? YES

Demonstrating an inhibitory effect of ESYT1/2 on T cell degranulation.

2. Is the reported work of significance (YES), or does it describe a confirmatory finding or one that has already been documented using other methods or in other organisms etc (NO)? YES, the report contains a substantial element of novelty in terms of the mechanistic insights into DAG.

3. Is it of general interest to the molecular biology community? YES, the report is of interest for the fields of T cell immunology and T cell-based immunotherapy.

4. Is the single major finding robustly documented using independent lines of experimental evidence (YES), or is it really just a preliminary report requiring significant further data to become convincing, and thus more suited to a longer format article (NO)? YES, the findings are robust with regard to the use of advanced analytical methods including cytotoxicity increase upon knockdown in primary CD8 T cells. However, the relevance of mechanistic claims for natural T cells is uncertain due to overreliance on the Jurkat cell

system for the mechanistic studies. Extending the mechanistic findings onto primary T cells with the use of MHC ligands for TCR activation would address this concern.

Answer: We appreciate the reviewer for the insightful comments, and we also agree that our original studies heavily relied on Jurkat cell lines. For these reason, in the revised version of the manuscript we incorporated data from primary CD8+ and CD4+ T cells on signaling (New Fig1, Fig. EV2), microscopy (new Fig. EV10) and IL2 production (New Fig3) positively correlate with those previously shown with Jurkat cell lines.

Additional points:

128: It is difficult to agree with the notion of E-Syt1 and E-Syt2 independency based on the finding that E-Syt1&2 double-knockdown did not produce an additive effect. A more fitting interpretation of this particular result would be that the roles are interdependent in a consecutive manner. If the roles were truly independent such as in disconnected parallel pathways, double knockdown would be expected to produce an additive effect.

Answer: We completely agree with the reviewer's comment . We have modified the text accordingly to reflect this interdependency roles of ESyt proteins.

267: Please clarify the sentence: "Given our previous findings that lack of expression of E-Syt2 alone in conjunction with E-Syt1 induces DAG accumulation at the PM, ..." Is it alone or in conjunction?

Answer: We apologize for this oversight, and we have corrected the text.

801: Phospho instead of phosphor.

Answer: We have now corrected this typo.

Figure legend titles for figures 2, 3, 4, 5 and extended figures 2 and 3 are misleading by stating T cells. The titles should clearly state Jurkat cells.

Answer: We apologize for this oversight; we have now corrected the titles with the correct terminology.

Figure 1 legend title should refer to both E-Syt1 and E-Syt2 (currently only ESyt1).

Answer: This is now corrected.

Fig 1C is difficult to decipher - please consider enlarging this graph.

Answer: We have modified this figure accordingly.

Fig 4 Color scheme is confusing by showing CD4 using the purple color for WT cells and the red color for KO cells. It would be better to use the same color. This experiment should not be divided into panels A and B given the groups are analyzed against each other (C, D).

Answer: We apologize for this oversight; we have maintained the same color scheme for CD4 (magenta) throughout all of Figure 4. Additionally, we have also added a WT

Jurkat cell in both conditions (resting and activated) to new Figure 4B (now showing pLAT instead of total LAT as suggested by Reviewer 2) panel with the rest of the Jurkat KO cell groups analyzed.

Dear Prof. Giraudo

Thank you for the submission of your revised manuscript to EMBO reports. We have now received the full set of referee reports that is copied below.

As you will see, both referees are very positive about the study and request only minor changes to clarify text, some experiments, and statistical analysis.

From the editorial side, there are also a few things that we need before we can proceed with the official acceptance of your study.

- Please provide up to 5 keywords on the title page.
- Please update the 'Conflict of interest' paragraph to our new 'Disclosure and competing interests statement'. For more information see <https://www.embopress.org/page/journal/14693178/authorguide#conflictsofinterest>
- References need to be alphabetical and et al should be used after 10 author names. DOIs should only be used for preprints and datasets that have not been published yet.
- Fig. 4C-D, and panels A, B of Fig. 6 are never called out in the text. Please add callouts where appropriate.
- Funding information discrepancy - grant numbers for HHS | NIH | National Institute of General Medical Sciences (NIGMS) and HHS | NIH | National Institute of Allergy and Infectious Diseases (NIAID) are listed in the manuscript, but the actual funder names are missing.
- Please note that we can only typeset up to 5 EV figures. Please choose 5 and provide the rest as Appendix. The Appendix is a single pdf file including all figures and their legends. We also need a title page with a table of content and page numbers. The nomenclature is Appendix Figure S# and the callouts in the manuscript need to be updated accordingly.
- Source data for the main figures need to be uploaded as one zipped folder per figure, each including subfolders for the individual panels. This is not needed for the source data that are deposited on BiImage. Folders for EV figures can be grouped and uploaded as one zipped folder.
- Source data for Figure 5 is partially mislabeled. 5A needs to be called 5B, and 5B needs to be called 5D.
- Data availability section: Please provide the accession codes and URLs for BiImage and BioStudies datasets. The links should directly resolve to the deposited datasets.
- Author checklist: please complete the third column (In which section is the information available?). Laboratory protocol (section Design) only applies to step-by-step protocols published in other journals or repositories.
- Please provide catalog or RRID numbers for the antibodies listed in the materials and methods section.
- As a standard procedure, we edit the title and abstract of manuscripts to make them more accessible to a general readership. Please find the edited versions below my signature and integrate the changes into the manuscript file if you agree with these.
- EMBO Reports papers are accompanied online by A) a short (1-2 sentences) summary of the findings and their significance, B) 2-3 bullet points highlighting key results and C) a synopsis image that is 550x300-600 pixels large (width x height) in PNG for JPG format. You can either show a model or key data in the synopsis image. Please note that the size is rather small and that text needs to be readable at the final size. Please send us this information along with the revised manuscript.
- Our production/data editors have asked you to clarify several points in the figure legends (see below). Please incorporate these changes in the manuscript and return the revised file with tracked changes with your final manuscript submission.

A) Figure legend text:

1. Please note that in figure EV2 specific figure legends are incorrectly labeled as ""b, c, d"" instead of ""a, b, c"".
2. Please note that in figures 5, EV5 legends, the panel descriptions do not follow alphabetic order.

B) Statistical test information (only p-values that are actually shown in the figure should (and must) be defined in the legends, all others should be removed (or added) from (to) the legend):

1. Please note that in figures 1b; 2f; 3d-e comparison for "****/**/*****" has not been represented in the figure panels.
2. Please note that in figures 1c; EV5d comparison for "*/****" has not been represented in the figure panels.
3. Please note that in figures 1d-f; 3b comparison for "****/*****" has not been represented in the figure panels.
4. Please note that in figures 5b, d comparison for "*****" has not been represented in the figure panels.
5. Please note that in figures EV 2a-c; EV8b-c; EV10b comparison for "*****" has not been represented in the figure panels.
6. Please note that in figures EV7b-c comparison for "*/****" has not been represented in the figure panels.
7. Please note that in Figure EV9a-b; EV11a-b comparison for "*/**/****/*****" has not been represented in the figure panels.
8. Please define the annotated p values ****/**/** in the legends of figures EV3b-d; EV4a-b
9. Please define the annotated p values **/* in the legends of figures 2b-c.
10. Please define the annotated p values **** in the legend of figure EV5b
11. Please indicate the statistical test used for data analysis in the legends of figures 2b-c; 3a, c; EV5b

C) Replicates and error bars:

1. Please note that the error bars are not defined in the legends of figures 1a; 2b-d; 3a, c; 5b, d; EV3b-g; EV5b
2. Please note that information related to n is missing in the legend of figures 1a; 2b-d; 3a, c

D) Data presentation/micrographs:

1. Please note that scale bar and its definition are missing for figures 4b; 5a, c; EV5a, c; EV6a-b; EV8a; EV10a
2. Please define what white arrows in figures 4a; EV7a.

- On a different note, I would like to alert you that EMBO Press offers a new format for a video-synopsis of work published with us, which essentially is a short, author-generated film explaining the core findings in hand drawings, and, as we believe, can be very useful to increase visibility of the work. This has proven to offer a nice opportunity for exposure i.p. for the first author(s) of the study. Please see the following link for representative examples and their integration into the article web page:

<https://www.embopress.org/doi/full/10.15252/embj.2019103932>

With kind regards,

Extended-Synaptotagmin-1 and -2 Control T cell signaling and function

Upon T cell activation, the levels of the second messenger diacylglycerol (DAG) at the plasma membrane need to be controlled to ensure appropriate T cell receptor signaling and T cell functions. Extended-Synaptotagmins (E-Syts) are a family of inter-organelle lipid transport proteins that bridge the endoplasmic reticulum and the plasma membrane. In this study, we identify a novel regulatory mechanism of DAG-mediated signaling for T cell effector functions based on E-Syt proteins. We demonstrate that E-Syts downmodulate T-cell receptor signaling, T cell-mediated cytotoxicity, degranulation, and cytokine production by reducing plasma membrane levels of DAG. Mechanistically, E-Syt2 predominantly modulates DAG levels at the plasma membrane in resting state T cells, while E-Syt1 and E-Syt2 negatively control T-cell receptor signaling upon stimulation. These results reveal a previously underappreciated role of E-Syts in regulating DAG dynamics in T cell signaling.

Referee #1:

The authors have addressed all the issues. The revised version is adequate for publication in EMBO reports

Referee #2:

Although the authors have answered appropriately most of my comments, I still have some concerns. Please find below my specific comments.

The manuscript "Extended-Synaptotagmin-1 and -2 Control Signaling and Functionality of T Cells at the Immune Synapse", is the first description of a novel role of E-Syts in DAG dynamics at the immune synapse. The concept that mechanisms of lipid regulation at the synaptic membrane via lipid transport play roles in TCR signaling is an interesting notion.

The data support the author's central finding. One important point that is not very well addressed is how E-Syts regulate DAG dynamics in T cells. It is not clear if the increase of DAG occurs at the expense of PI(4,5)P2 cleavage or is mediated by transport of DAG. Are diacylglycerol kinases or other molecules involved in this process? In addition, it could be very interesting to see ER-PM contact sites at the IS.

Answer: We appreciate these critical insights. To address this question we now included data from new experiments using either PLC γ or DGK inhibitors showing that the accumulation of DAG at the PM of ESyt-KO cells depends upon PLC γ activity (Fig EV 11, and 12). Interestingly, the addition of DGK inhibitor did not alter DAG levels in resting conditions but increased DAG at the PM in ESyt2-KO-activated cells. These data support a model in which ESyts control DAG levels at resting state while both ESyts and DGKs modulates DAG levels in activated conditions.

-Figure EV11 and 12: please check the reference to the figure in the main text (line 333, 336 and 340). Are the results in figures EV 11 and 12 statistically significant?

Line 115: "Knockdown efficiency was tested by WB", please indicate the quantification of KD efficiency.

Answer: we apologize for not adding this information before. We have added the quantifications of KD efficiency in the new Fig 1a.

-Figure 1A: the graph showing the quantification of KD is not clear, please modify. Please add the statistical analysis.

Lines 111-130: it could be interesting to analyse the polarization of lytic granules and the localization of DAG at the lytic synapse. Is statistically significant the increase in degranulation in unstimulated KD CD8+ T cells showed in figure 1B?

Answer: We appreciate this interesting comment. To address this, we have performed new imaging studies using ESyt KD human PBMCs and analyzed the colocalization of the lytic granule marker CD63 with DAG at the PM in either resting or activated conditions. These results revealed an increased colocalization of CD63 in ESYT KD cells compared with WT cells. This data is now included in new Fig. EV10.

-Why the authors use PBMCs and CD63 instead of CD8+ cells and LAMP1, granzyme or perforin as lytic granule marker? CD63 is expressed on cell surface and granular membranes of many hematopoietic cells: it is not sure that the authors are looking T cells. I suggest removing these results in the final version of the manuscript. Note that in the figure legend the authors indicate Jurkat instead of PBMCs.

Lines 296-298: "Unlike our previous results in resting state conditions which showed that only E-Syt2 KO cells displayed a spike in DAG accumulation within the first few minutes of contact with the surface", this is not clear from the showed graph. Why DAG levels are higher in resting than in activated E-Syt2KD cells?

Answer: To clarify this issue, we want to point out that the quantification in the graph shows the DAG fluorescent intensity ratio between the center of the contact point area and the cell periphery. In resting ESyt KO cells, the amount of DAG in the cell periphery is relatively low compared with the center, whereas in the activated conditions the total DAG levels significantly increase in both the center and periphery, thus resulting in a lower ratio. As mentioned above we have added the quantification of the total DAG measured as total integrated density (Fig. 5), showing a higher level of DAG in activated vs resting.

-Please perform the statistical analysis of these results. Is the increase in apoptosis observed in activated E-Syt1 KO cells not significant?

Line 367: "We observed that in the absence of E-Syts, cells bypass TCR-activation and are in a pre-activated state, independent of standard stimulation". I am wondering if there is an induction of apoptosis after T cell stimulation (AICD, activation induced cell death). Is cell viability modulated by the expression of E-Syt proteins?

Answer: We thank the reviewer for bringing up this important point. To address this question we performed an experiment to measure activation induced cell-death in WT and ESyt KO cell lines using a flow cytometry assay. Results from these experiments showed that ESyt KO cell lines did not show any statistically significant difference in cell death upon activation when compared with WT-cells. This data is now included in new Fig EV. 13.

-Please add the statistical analysis of these results. Is the increase in apoptosis observed in activated E-Syt1 KO cells not significant?

Response to the Reviewers and Editorial Office

Referee #1:

The authors have addressed all the issues. The revised version is adequate for publication in EMBO reports

Referee #2:

Although the authors have answered appropriately most of my comments, I still have some concerns. Please find below my specific comments.

The manuscript "Extended-Synaptotagmin-1 and -2 Control Signaling and Functionality of T Cells at the Immune Synapse", is the first description of a novel role of E-Syts in DAG dynamics at the immune synapse. The concept that mechanisms of lipid regulation at the synaptic membrane via lipid transport play roles in TCR signaling is an interesting notion. The data support the author's central finding. One important point that is not very well addressed is how E-Syts regulate DAG dynamics in T cells. It is not clear if the increase of DAG occurs at the expense of PI(4,5)P2 cleavage or is mediated by transport of DAG. Are diacylglycerol kinases or other molecules involved in this process? In addition, it could be very interesting to see ER-PM contact sites at the IS.

Answer: We appreciate these critical insights. To address this question we now included data from new experiments using either PLC γ or DGK inhibitors showing that the accumulation of DAG at the PM of E-syt-KO cells depends upon PLC γ activity (Fig EV 11, and 12). Interestingly, the addition of DGK inhibitor did not alter DAG levels in resting conditions but increased DAG at the PM in E-syt2-KO-activated cells. These data support a model in which E-syts control DAG levels at resting state while both E-syts and DGKs modulates DAG levels in activated conditions.

-Figure EV11 and 12: please check the reference to the figure in the main text (line 333, 336 and 340). Are the results in figures EV 11 and 12 statistically significant?

Answer: We apologize for not including the statistical analysis on these results. We have revised both figures (now EV4 and EV5 respectively)

We would like to point out that for the data in Fig EV11 (now EV4), we only included the statistical analysis results at 5-time points in each graph for simplicity. Each time point is 50 seconds apart from one another. As stated in the figure legend, we carried out comparisons against respective non-treated cell line controls.

For Fig EV12 (now EV5) we carried out statistical analysis comparisons against the respective non-treated cell line control.

Line 115: "Knockdown efficiency was tested by WB", please indicate the quantification of KD efficiency.

Answer: we apologize for not adding this information before. We have added the quantifications of KD efficiency in the new Fig 1a.

-Figure 1A: the graph showing the quantification of KD is not clear, please modify. Please add the statistical analysis.

Answer: We apologize for the lack of clarity with this graph. We have revised this graph to indicate in light gray protein expression of E-Syt1 and E-Syt2 in darker gray. We have added the statistical analysis for this data on the graph and further explained on the Figure legend.

Lines 111-130: it could be interesting to see the polarization of lytic granules and the localization of DAG at the lytic synapse. Is statistically significant the increase in degranulation in unstimulated KD CD8+ T cells showed in figure 1B?

Answer: We appreciate this interesting comment. To address this, we have performed new imaging studies using E-syt KD human PBMCs and analyzed the colocalization of the lytic granule marker CD63 with DAG at the PM in either resting or activated conditions. These results revealed an increased colocalization of CD63 in ESYT KD cells compared with WT cells. This data is now included in new Fig. EV10.

-Why the authors use PBMCs and CD63 instead of CD8+ cells and LAMP1, granzyme or perforin as lytic granule marker? CD63 is expressed on cell surface and granular membranes of many hematopoietic cells: it is not sure that the authors are looking T cells. I suggest removing these results in the final version of the manuscript. Note that in the figure legend the authors indicate Jurkat instead of PBMCs.

Answer: We want to point out that we used PBMCs expanded with CD3/28 beads for at least 8 days, which resulted in population of over 98% of CD4⁺ or CD8⁺ T cells in culture, ruling out the presence of other immune cells. Regarding the use of CD63 as lytic granule localization marker, there are several publications that show that CD63 colocalizes with perforin-containing granules in T cells [Stichombe et. al. (2006) Nature; Blott et. al. (2002) Nat. Rev. Mol. Cell Biol]. Therefore we used it as a surrogate marker for lytic granules. We have also corrected the legend figure. We apologize for the mislabeling on the figure legend and have corrected this on the revised manuscript.

Lines 296-298: "Unlike our previous results in resting state conditions which showed that only E-Syt2 KO cells displayed a spike in DAG accumulation within the first few minutes of contact with the surface", this is not clear from the showed graph. Why DAG levels are higher in resting than in activated E-Syt2KD cells?

Answer: To clarify this issue, we want to point out that the quantification in the graph shows the DAG fluorescent intensity ratio between the center of the contact point

area and the cell periphery. In resting Esyt KO cells, the amount of DAG in the cell periphery is relatively low compared with the center, whereas in the activated conditions the total DAG levels significantly increase in both the center and periphery, thus resulting in a lower ratio. As mentioned above we have added the quantification of the total DAG measured as total integrated density (Fig. 5), showing a higher level of DAG in activated vs resting.

-Please perform the statistical analysis of these results. Is the increase in apoptosis observed in activated E-Syt1 KO cells not significant?

***Answer:** We don't fully understand if the reviewer was referring to data from EV13 (Apoptosis Assay), rather than Figure 5 as noted in the previous comments. Nonetheless, we included the statistical analysis for this figure looking at apoptosis (EV13) and it is now on Appendix Figure S8.*

Line 367: "We observed that in the absence of E-Syts, cells bypass TCR-activation and are in a pre-activated state, independent of standard stimulation". I am wondering if there is an induction of apoptosis after T cell stimulation (AICD, activation induced cell death). Is cell viability modulated by the expression of E-Syt proteins?

Answer: We thank the reviewer for bringing up this important point. To address this question we performed an experiment to measure activation induced cell-death in WT and Esyt KO cell lines using a flow cytometry assay. Results from these experiments showed that Esyt KO cell lines did not show any statistically significant difference in cell death upon activation when compared with WT-cells. This data is now included in new Fig EV. 13.

-Please add the statistical analysis of these results. Is the increase in apoptosis observed in activated E-Syt1 KO cells not significant?

***Answer:** We apologize for not including the statistical analysis on these results. We have added the results to the now new Appendix Figure S8 and we indicate that upon analysis of all the cell lines in resting or activated conditions respectively, we saw a statistically significant increase in Annexin V in E-Syt1 KO-activated cells when compared to their WT counterparts. Despite this increase, we did not observe any impact on this cell line's growth throughout our experiments however, this is a point that could be further looked at in future studies.*

Editorial Office notes:

From the editorial side, there are also a few things that we need before we can proceed with the official acceptance of your study.

- Please provide up to 5 keywords on the title page.

***Answer:** Extended-Synaptotagmins, T cells, plasma membrane, DAG, T-cell receptor signaling*

- Please update the 'Conflict of interest' paragraph to our new 'Disclosure and competing interests statement'. For more information see <https://www.embopress.org/page/journal/14693178/authorguide#conflictsofinterest>

Answer: *We have modified the text according to the journal guidance.*

- References need to be alphabetical and et al should be used after 10 author names. DOIs should only be used for preprints and datasets that have not been published yet.

Answer: *We have modified the reference list accordingly.*

- Fig. 4C-D, and panels A, B of Fig. 6 are never called out in the text. Please add callouts where appropriate.

Answer: *We have modified the text and added the called out for those figures.*

- Funding information discrepancy - grant numbers for HHS | NIH | National Institute of General Medical Sciences (NIGMS) and HHS | NIH | National Institute of Allergy and Infectious Diseases (NIAID) are listed in the manuscript, but the actual funder names are missing.

Answer: *We have added the funding institution information .*

- Please note that we can only typeset up to 5 EV figures. Please choose 5 and provide the rest as Appendix. The Appendix is a single pdf file including all figures and their legends. We also need a title page with a table of content and page numbers. The nomenclature is Appendix Figure S# and the callouts in the manuscript need to be updated accordingly.

Answer: *To comply with the journal requirements we have kept old EV Figures 2, 5, 8, 11 and 12 and organized the rest of the EV Figures 1 ,3, 4, 6, 7, 9, 10 and 13 as Appendix Figures.*

- Source data for the main figures need to be uploaded as one zipped folder per figure, each including subfolders for the individual panels. This is not needed for the source data that are deposited on BioImage. Folders for EV figures can be grouped and uploaded as one zipped folder.

Answer: *We have uploaded all the source data of the main figures a individual zip files and grouped all the EV figures source data in single zip-file.*

- Source data for Figure 5 is partially mislabeled. 5A needs to be called 5B, and 5B needs to be called 5D.

Answer: *We have corrected the labeling of the folder of those figures.*

- Data availability section: Please provide the accession codes and URLs for BioImage and BioStudies datasets. The links should directly resolve to the deposited datasets.

Answer: *Since we have been able to upload all the source data figures files, including the images, this section does not apply anymore.*

- Author checklist: please complete the third column (In which section is the information available?). Laboratory protocol (section Design) only applies to step-by-step protocols

published in other journals or repositories.

Answer: *We have updated this file accordingly.*

- Please provide catalog or RRID numbers for the antibodies listed in the materials and methods section. *

Answer: *We have added this information in the text.*

- As a standard procedure, we edit the title and abstract of manuscripts to make them more accessible to a general readership. Please find the edited versions below my signature and integrate the changes into the manuscript file if you agree with these.

Answer: *We agree with these changes and we have replaced the title and abstract with new version provided by the journal.*

- EMBO Reports papers are accompanied online by A) a short (1-2 sentences) summary of the findings and their significance, B) 2-3 bullet points highlighting key results and C) a synopsis image that is 550x300-600 pixels large (width x height) in PNG for JPG format. You can either show a model or key data in the synopsis image. Please note that the size is rather small and that text needs to be readable at the final size. Please send us this information along with the revised manuscript.

Answer:

A) Short summary

Extended-Synaptotagmin1 and -2 (E-Syt1 and -2) are lipid transport proteins located at the interphase of ER-plasma membrane contact sites and are responsible for downmodulating DAG levels at the plasma membrane, T-cell receptor signaling, and T-cell effector functions.

B) -2-3 bullet points

- *E-Syt proteins regulate T-Cell receptor signaling and T-cells functions through downmodulating DAG levels at the plasma membrane.*
- *E-Syt2 predominantly modulates plasma membrane DAG levels in resting state T cells.*
- *E-Syts proteins redistribute into DAG-rich regions upon T-cell activation.*

C) Synopsis image created and provided as attachment file

- Our production/data editors have asked you to clarify several points in the figure legends (see below). Please incorporate these changes in the manuscript and return the revised file with tracked changes with your final manuscript submission.

A) Figure legend text:

1. Please note that in figure EV2 specific figure legends are incorrectly labeled as ""b, c, d"" instead of ""a, b, c"".
2. Please note that in figures 5, EV5 legends, the panel descriptions do not follow alphabetic order.

Answer: *We have corrected all these noted issues.*

B) Statistical test information (**only p-values that are actually shown in the figure should**

(and must) be defined in the legends, all others should be removed (or added) from (to) the legend):

1. Please note that in figures 1b; 2f; 3d-e comparison for ""***/***/****"" has not been represented in the figure panels.
2. Please note that in figures 1c; EV5d comparison for ""*/****"" has not been represented in the figure panels.
3. Please note that in figures 1d-f; 3b comparison for ""****/****"" has not been represented in the figure panels.
4. Please note that in figures 5b, d comparison for ""****"" has not been represented in the figure panels.
5. Please note that in figures EV 2a-c; EV8b-c; EV10b comparison for ""****"" has not been represented in the figure panels.
6. Please note that in figures EV7b-c comparison for ""*/****"" has not been represented in the figure panels.
7. Please note that in Figure EV9a-b; EV11a-b comparison for ""*/**/***/****"" has not been represented in the figure panels.
8. Please define the annotated p values *****/***/**/* in the legends of figures EV3b-d; EV4a-b
9. Please define the annotated p values **/* in the legends of figures 2b-c.
10. Please define the annotated p values **** in the legend of figure EV5b
11. Please indicate the statistical test used for data analysis in the legends of figures 2b-c; 3a, c; EV5b

Answer: *We have corrected all these noted issues.*

C) Replicates and error bars:

1. Please note that the error bars are not defined in the legends of figures 1a; 2b-d; 3a, c; 5b, d; EV3b-g; EV5b
2. Please note that information related to n is missing in the legend of figures 1a; 2b-d; 3a, c

Answer: *We have corrected all these noted issues*

D) Data presentation/micrographs:

1. Please note that scale bar and its definition are missing for figures 4b; 5a, c; EV5a, c; EV6a-b; EV8a; EV10a
2. Please define what white arrows in figures 4a; EV7a.

Answer: *We have corrected all these issues noted by the production department.*

Prof. Claudio Giraud
Thomas Jefferson University
Microbiology and Immunology
233 S 10th Street
BLSB 608
Philadelphia, Pennsylvania 19107
United States

Dear Prof. Giraud,

I am very pleased to accept your manuscript for publication in the next available issue of EMBO reports. Thank you for your contribution to our journal.

Yours sincerely,
